# DO GENERATED DATA ALWAYS HELP CONTRASTIVE LEARNING?

**Yifei Wang**[1]* **Jizhe Zhang**[2]* **Yisen Wang**[3,4]†

[1] School of Mathematical Sciences, Peking University
[2] Institute of Artificial Intelligence and Robotics, Xi'an Jiaotong University
[3] National Key Lab of General Artificial Intelligence,
  School of Intelligence Science and Technology, Peking University
[4] Institute for Artificial Intelligence, Peking University

## ABSTRACT

Contrastive Learning (CL) has emerged as one of the most successful paradigms for unsupervised visual representation learning, yet it often depends on intensive manual data augmentations. With the rise of generative models, especially diffusion models, the ability to generate realistic images close to the real data distribution has been well recognized. These generated high-equality images have been successfully applied to enhance contrastive representation learning, a technique termed "data inflation". However, we find that the generated data (even from a good diffusion model like DDPM) may sometimes even harm contrastive learning. We investigate the causes behind this failure from the perspective of both data inflation and data augmentation. For the first time, we reveal the complementary roles that stronger data inflation should be accompanied by weaker augmentations, and vice versa. We also provide rigorous theoretical explanations for these phenomena via deriving its generalization bounds under data inflation. Drawing from these insights, we propose **Adaptive Inflation (AdaInf)**, a purely data-centric strategy without introducing any extra computation cost. On benchmark datasets, AdaInf can bring significant improvements for various contrastive learning methods. Notably, without using external data, AdaInf obtains 94.70% linear accuracy on CIFAR-10 with SimCLR, setting a new record that surpasses many sophisticated methods. Code is available at `https://github.com/PKU-ML/adainf`.

## 1 INTRODUCTION

Contrastive learning has to be the state-of-the-art method for self-supervised representation learning across many domains (Chen et al., 2020a; He et al., 2020; Wang et al., 2021a; Guo et al., 2023; Zhang et al., 2023a;b). However, there still remains a noticeable gap in performance compared to its supervised counterparts (Chen et al., 2021; Cui et al., 2023). Among many attempts to close this gap, a recent surge of interest lies in leveraging high-quality generative models to boost contrastive learning (Wu et al., 2023; Wang et al., 2022a; Azizi et al., 2023; Tian et al., 2023). Given an unlabeled dataset, *e.g.,* CIFAR-10, one can train a generative model on it (*e.g.,* DDPM (Ho et al., 2020)) to generate a lot of synthetic samples, and then perform contrastive learning on the combination of the real and generated data. This simplest way for using generated data is called "*data inflation*". Notably, it is orthogonal to the *data augmentation* process, wherein an image–be it raw or generated–is subjected to manual augmentations (*e.g.,* cropping) to yield positive and negative pairs used in contrastive learning (see Figure 1(a) for an illustration of the pipeline).

Though one may expect that generated data will benefit contrastive learning with more diverse data, we find that it is *not* always the case. As shown in Figure 1(b), simply inflating CIFAR-10 with 1M images generated by DDPM (the vanilla setting) leads to even worse linear probing accuracy (91.33% *v.s.* 90.27%). We investigate this unexpected performance degradation from two aspects: data inflation (how inflated data are constructed) and data augmentation (how to craft augmented

---

*Equal Contribution. Yifei Wang has graduated from Peking University, and is currently a postdoc at MIT.
†Corresponding Author: Yisen Wang (yisen.wang@pku.edu.cn).

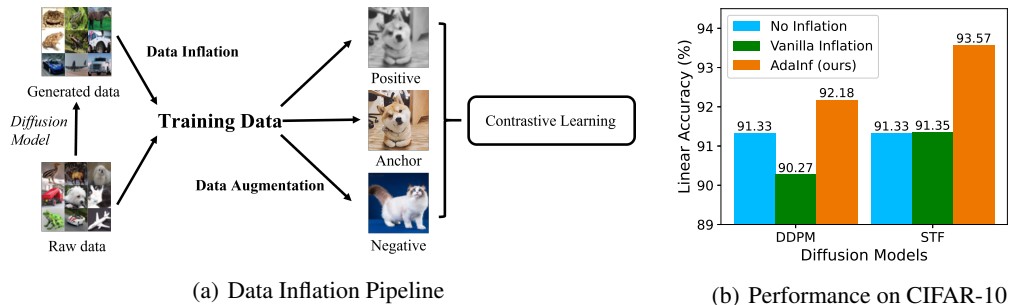

(a) Data Inflation Pipeline   (b) Performance on CIFAR-10

Figure 1: (a): During data inflation, the real data and the generated data (usually with a larger size) are combined together as the training data for contrastive learning, where two random augmentations are drawn from each sample to compute the contrastive loss. (b): Linear accuracy of contrastive learning (Chen et al., 2020a) with different data inflation strategies on CIFAR-10. The generated data are 1M samples drawn from DDPM (with 3.04 FID) or STF (with 1.94 FID).

samples with inflated data). For the former, we find that better generation quality is of limited help, while reweighting real and generated data can attain larger gains. For the latter, we discover an intriguing phenomenon that weaker data augmentation, although harmful in standard contrastive learning (Tian et al., 2020; Luo et al., 2023), can be very helpful with data inflation. To uncover the mysteries behind these observations, we establish the first generalization guarantees for inflated contrastive learning, and explain the benefits of weak augmentations by revealing the complementary roles between data inflation and data augmentation. Based on these insights, we propose an Adaptive Inflation (AdaInf) strategy that adaptively adjusts data augmentation strength and mixing ratio for data inflation, which can bring significant improvements in downstream performance without any introducing computation overhead. We summarize our contributions as follows:

- We discover a failure mode of data inflation for contrastive learning, and reveal the causes of this failure from the perspective of both data inflation and data augmentation. In particular, we find that data reweighting and weak augmentation contribute significantly to improving final performance.

- To understand these phenomena, we establish the first theoretical guarantees for inflated contrastive learning, which not only rigorously explain previous phenomena, but also reveal the complementary roles between data inflation and data augmentation.

- We propose the Adaptive Inflation (AdaInf) strategy to adaptively adjust data augmentation strength and mixing ratio for data inflation. Extensive experiments show that the proposed approach improves downstream accuracy significantly at no extra cost, and it is particularly beneficial for data-scarce scenarios.

## 2 PRELIMINARY & RELATED WORK

**Self-Supervised Learning.** Given an unlabelled dataset $D$ that contain raw samples $\bar{x} \in \mathbb{R}^d$, the goal of self-supervised learning is to pretrain a feature extraction $f : \mathbb{R}^d \to \mathbb{R}^z$ on $\mathcal{D}$ such that the learned representations generalize well to downstream tasks. In this paper, we mainly consider contrastive learning as a representative example. For each sample $\bar{x} \in \mathcal{D}$, we draw two randomly augmented samples $x, x^+ \sim \mathcal{A}(\cdot|\bar{x})$ as a positive pair. The general learning objective of contrastive learning is to align the representations of positive samples while pushing negative samples apart, as in the following widely adopted InfoNCE loss (Oord et al., 2018; Wang and Isola, 2020):

$$\mathcal{L}_{\text{InfoNCE}}(f, \mathcal{D}) = -\mathbb{E}_{x,x^+,\{x_i^-\}_{i=1}^M} \log \frac{\exp\left(f(x)^\top f(x^+)\right)}{\exp\left(f(x)^\top f(x^+)\right) + \sum_{i=1}^M \exp\left(f(x)^\top f(x_i^-)\right)}, \quad (1)$$

where $\{x_i^-\}_{i=1}^M$ are $M$ negative samples drawn independently from $D$ with data augmentation $\mathcal{A}(\cdot)$. Besides, some variants propose to drop negative samples and adopt asymmetric modules to encoder positive pairs to avoid feature collapses (Grill et al., 2020; Chen and He, 2021; Caron et al., 2020;

2021; Zhuo et al., 2023). Some propose to use regularization terms to replace the negative samples and obtain similar performance (Zbontar et al., 2021). Recent theoretical analyses show there exists a deep connection between contrastive learning and these variants (Tian et al., 2021; Garrido et al., 2023; Wang et al., 2023). Therefore, we regard them as general contrastive learning methods.

**Generative Models.** Generative models refer to a broad class of models that learn the data distribution $P(x)$. Popular generative models include GANs (Goodfellow et al., 2014; Wang et al., 2021b), VAEs (Kingma and Welling, 2014), diffusion models (Ho et al., 2020), *etc.* In this paper, we mainly take diffusion models for an example due their superior generation quality. During training time, we add random Gaussian noise of scale $t \in [0, T]$ to an image $\bar{x} \in \mathcal{D}$, and train a denoising network $\varepsilon_\theta : \mathcal{R}^d \to \mathcal{R}^d$ (typically a U-Net) to reconstruct the ground-truth noise added to the image $\bar{x}$, *i.e.,*

$$\mathcal{L}_{SM}(g, \mathcal{D}) = \mathbb{E}_{t, \bar{x} \in \mathcal{D}, \varepsilon_t} \| \varepsilon_\theta(\sqrt{\bar{\alpha}_t}\bar{x} + \sqrt{1 - \bar{\alpha}_t}\varepsilon_t, t) - \varepsilon_t \|^2, \tag{2}$$

where $\bar{\alpha}_t$ is the mixing coefficient at time $t$ (Ho et al., 2020). In this paper, to enhance contrastive learning on unlabeled data, we train an unsupervised diffusion model with real data (*e.g.,* CIFAR-10), sample one million generated samples from the diffusion model, and append them to the real data. During this process, we can inflate the training data from 50k samples to more than 1M samples, so we call it data inflation. Remarkably, we **do not use any external data or model** since the diffusion model is also trained on the same training dataset. See Appendix A for more related work on learning with generated data.

## 3 UNCOVERING REASONS BEHIND THE FAILURE OF DATA INFLATION

As shown in Figure 1(b), we discover that directly adding 1M images generated by DDPM (Ho et al., 2020) may yield minimal or even negative improvements on contrastive learning. In this section, we explore the reasons behind this failure from two aspects, the generated data and data augmentation, and design effective strategies to mitigate these failures.

### 3.1 CAUSES IN DATA INFLATION: DATA QUALITY AND DATA REWEIGHTING

First, we investigate whether the failure lies in our design of data inflation. Denote the distribution of the real data $\mathcal{D}_d$ as $P_d$, and that of the generated data $\mathcal{D}_g$ as $P_g$. After inflation, the overall training distribution becomes $P_t = \beta P_d + (1 - \beta)P_g$, where $\beta = |\mathcal{D}_d|/(|\mathcal{D}_d| + |\mathcal{D}_g|)$ denotes the proportion of the real data when *equally* mixing them together. The distribution gap between real and generated data can be characterized by the following Theorem 3.1 (proof in Appendix D.1):

**Theorem 3.1.** $\mathrm{D}_{\mathrm{TV}}(P_t, P_d) = (1 - \beta) \, \mathrm{D}_{\mathrm{TV}}(P_g, P_d)$, *where* $\mathrm{D}_{\mathrm{TV}}$ *denotes the TV distance.*

From the above, we can see that there are two factors influencing the distribution gap: the generated data $P_g$ and the mixing ratio $\beta$.

**Generated Data Quality.** A straightforward reason for the failure is that the generative model, DDPM, is not good enough. Since the generative model is not perfect, the distribution gap between real and generated data will be large. Thus, there will be a large mismatch between training and testing data, which prevents generalization. In turn, as long as $P_g = P_d$, generated data will always be helpful (with more training examples). Thus, a direct solution to the degradation is to use a better generative model with a smaller gap to real data. To validate this point, we compare four diffusion models with different generation qualities (measured by FID). Figure 2(a) shows that indeed, diffusion models with lower FID, such as STF (Xu et al., 2023), consistently bring better downstream accuracy. However, we also notice two drawbacks. First, better generative quality often requires larger models and/or slower sampling (*e.g.,* more denoising steps) (Bond-Taylor et al., 2022), which detriments the efficiency. Second, the improvement over the baseline (91.33%) is marginal, as using the best diffusion model STF only gains +0.02% accuracy, which is less worth the effort. Therefore, in the rest of the discussion, we fix the generative model to be STF and explore how to improve downstream performance with other techniques.

**Data Reweighting.** Beside generated data quality, Theorem 3.1 suggests another useful strategy, data reweighting. We can upweight the real data (or downweighting the generated data) with a larger mixing ratio $\beta$ which can lead to a smaller gap $\mathrm{D}_{\mathrm{TV}}(P_t, P_d)$. In practice, we upweight the real data by replicating it $N$ times during the mixing (equivalent to $\beta = N \cdot |\mathcal{D}_d|/(N \cdot |\mathcal{D}_d| + |\mathcal{D}_g|)$).

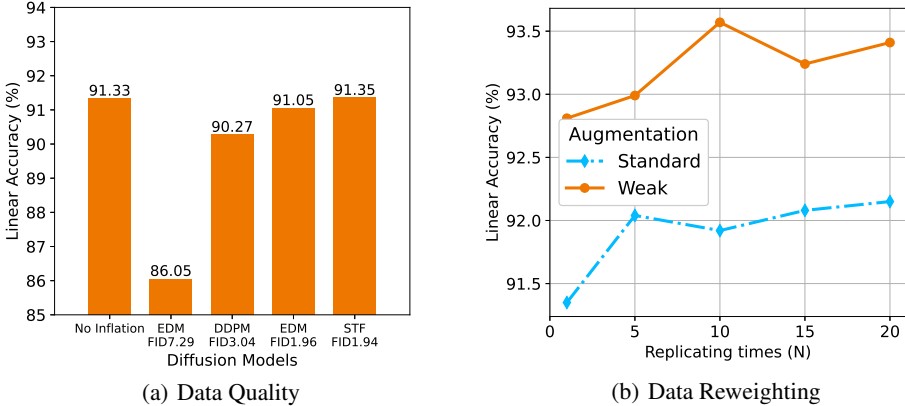

(a) Data Quality

(b) Data Reweighting

Figure 2: Performance of contrastive learning with 1M generated data on CIFAR-10. (a): Linear accuracy using four diffusion models for generation: DDPM (Ho et al., 2020), EDM (Karras et al., 2022), STF (Xu et al., 2023) (two EDM models differ in their training time). (b): Linear accuracy with data reweighting (real data: generative data $= N : 1$).

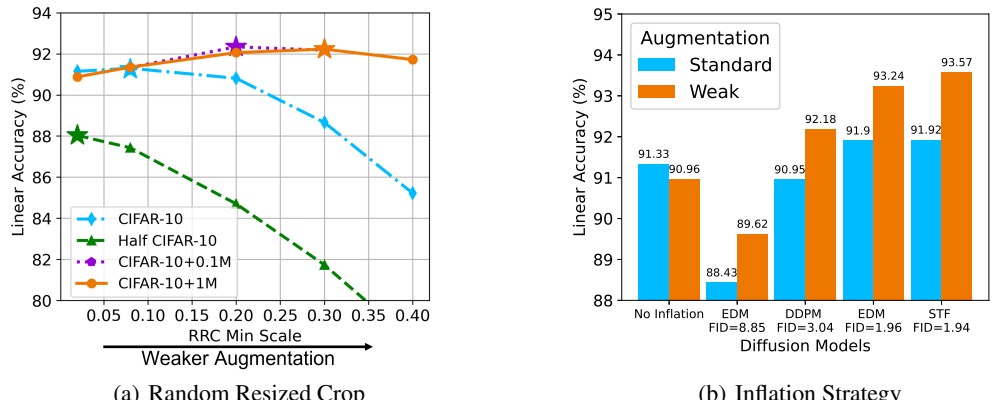

(a) Random Resized Crop

(b) Inflation Strategy

Figure 3: (a): Linear accuracy with different augmentation strengths, by changing the min scale of random resized cropping (lower value represents stronger augmentation). (b): Linear accuracy of different inflation strategies on CIFAR-10 (with $10 : 1$ data reweighting).

Figure 2(b) shows that after data reweighting, the linear accuracy reaches 92.15 % ($N = 20$) under standard augmentation, indicating that a real sample is worth multiple generated samples. Notably, more replication beyond $N = 10$ leads to worse performance under weak augmentation, as the low weight of generated data (with even higher replication of real data) now hinders the benefits in data diversity brought by generated data.

## 3.2 CAUSES IN DATA AUGMENTATION

Aside from the data inflation strategy, we also wonder whether the current training protocol of contrastive learning should also be adjusted for a much larger size of data (20x bigger). Data augmentation is arguably the most important part of contrastive learning (Wang et al., 2022b). The seminal work SimCLR (Chen et al., 2020a) shows that different data augmentations dramatically influence performance (much larger than learning objectives). Therefore, we might wonder how different choices of data augmentation affect the performance of data inflation.

Among commonly used augmentations, random resized crop (RRC) is the most important one (Chen et al., 2020a). Therefore, we adjust the augmentation strength by varying the minimal cropped (relative) area size, denoted as $a$ (by default, $a = 0.08$), and keeping others fixed. Smaller $a$ indicates a stronger augmentation that can crop the image to a smaller size, and vice versa. We compare four scale of training data: CIFAR-10, Half CIFAR-10 (50% random split), CIFAR-10 + 0.1M generated

data (with STF), and CIFAR-10 + 1M generated data (with STF). Figure 3(a) shows a clear trend that the optimal augmentation (marked by $\star$) is consistently weaker for larger training data (0.02 for Half CIFAR-10, 0.08 for CIFAR-10, 0.20 for CIFAR-10 + 0.1M, and 0.30 for CIFAR-10 + 1M). Therefore, more training data (especially with data inflation) requires an adaptive adjustment of augmentation strength to fully unleash its benefits. Guided by this principle, we propose a weak version of data augmentation (detailed in Section 5), and Figure 3(b) shows that this weak augmentation can consistently bring significant gains for generative data of different FIDs.

### 3.3 Proposed Strategy: Adaptive Inflation

Following these lessons so far, we arrive at a general guideline for data inflation: 1) we should put different weights on real and generated data, and worse quality data should have lower weights; 2) we should adopt milder data augmentations with more data. We call it Adaptive Inflation (**AdaInf**) to emphasize that we should adapt the training configurations according to the quality and size of generated data. In practice, to avoid exhaustive search, we adopt a default choice (called Simple AdaInf, or AdaInf for short) with $10 : 1$ mixture of real and generated data (Figure 2(b) shows that $10 : 1$ yields the best linear accuracy under weak augmentation.), and a weak data augmentation strategy designed following the AdaInf principle (details in Section 5). This default choice, as a baseline strategy for AdaInf training, works surprisingly well across multiple datasets and generated data. A preview performance of simple AdaInf is shown in Figure 1(b). With no downstream data, one may rely on surrogate metrics for finding the adaptive strategy (*e.g.,* ARC Wang et al. (2022b)).

## 4 Theoretical Characterization of Data Inflation

In Section 3, we show that different strategies in data inflation have large effects on downstream performance. In this section, we provide in-depth theoretical explanations of these phenomena.

### 4.1 Mathematical Formulation

To analyze the influence of data augmentation, we adopt the standard *augmentation graph* framework (HaoChen et al., 2021; Wang et al., 2022b), where data augmentations induce interactions (as edges) between training samples (as nodes). Different from their original setting dealing with in-domain generalization on population distribution, now we need to characterize the influence of adopting more training data and mismatched training-test distribution on downstream generalization.

**Raw Data as Subgraph.** To describe the difference between using raw data and inflated data, our key insight is that when the two have the same population distribution (perfect generation), the raw data can be seen as a random subset of the inflated data. This allows us to analyze their difference through a subsampled graph perspective in the augmentation graph framework. Denote the inflated dataset as $\bar{\mathcal{X}}$ and its augmented dataset as $\mathcal{X}$. We can define an augmentation graph over all augmented training samples in $\mathcal{X}$, and its adjacency matrix $A \in \mathbb{R}^{n \times n}$ represents the joint probability of positive samples under data augmentation, $A_{x,x'} = \mathbb{E}_{\bar{x} \sim \mathcal{P}_{\bar{\mathcal{X}}}} \mathcal{A}(x|\bar{x}) \mathcal{A}(x'|\bar{x})$. The (normalized) graph Laplacian is $L = I - D^{-\frac{1}{2}} A D^{-\frac{1}{2}}$, where $D$ is a diagonal degree matrix with the $(x,x)$-th diagonal element as $D_{xx} = \sum_{x'} A_{x,x'}$. Denote the eigenvalues of $\mathcal{L}$ as $0 = \lambda_1 \leq \lambda_2 \leq \cdots \leq \lambda_N \leq 2$. Ignoring the difference between real and generated data, we can regard the raw dataset $\bar{\mathcal{X}}_{\text{raw}}$ (real data) as a random subset of $\bar{\mathcal{X}}$ and accordingly, the augmentation graph of raw data is a random subgraph of $A$. This view allows us to use random graph theory to characterize the influence of data inflation.

Following common practice, we evaluate learned features with linear probing as the downstream task, where we learn a linear classifier with weights $B \in \mathbb{R}^{k \times r}$ ($r$ denotes the number of classes) as $g_{f,B}$ on top of pretrained features $f(x) \in \mathbb{R}^k$ to predict the labels $y \in \mathcal{Y}$ of augmented data. Then we define a majority voting classifier $\bar{g}_{f,B}(\bar{x}) := \arg\max_{i \in [r]} \Pr_{x \sim \mathcal{A}(\cdot|\bar{x})} (g_{f,B}(x) = i)$ to predict real data. A smaller classification error, denoted as $\mathcal{E}(f, B)$, indicates better feature separability.

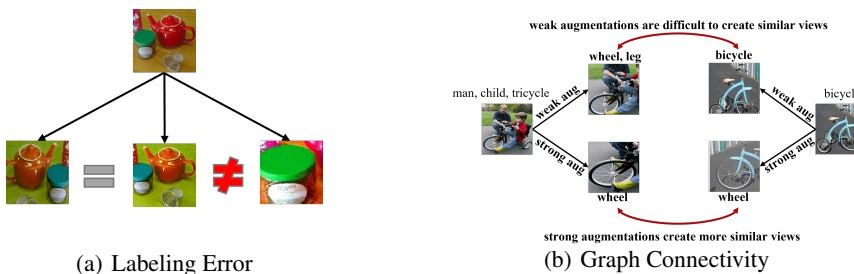

(a) Labeling Error        (b) Graph Connectivity

Figure 4: Illustrative examples on the effect of data augmentation on the labeling error ((a)) (*i.e.,* augmented samples belonging to different classes) and graph connectivity ((b)) (*i.e.,* different samples bridged together after augmentation).

## 4.2 GUARANTEES ON INFLATED CONTRASTIVE LEARNING

Given the formulation above, we establish a formal guarantee for contrastive learning with data inflation. Compared to the original result in HaoChen et al. (2021), our guarantees accommodate the discrepancies between the pretraining a downstream distribution (*i.e.,* OOD generalization).

**Theorem 4.1.** *With probability at least* $1 - \delta$, *for the optimal encoder* $f^*$ *on inflated data and a learned linear head* $B^*$, *its linear probing error has the following upper bound,*

$$\mathcal{E}(f^*, B^*) \leq \frac{8\alpha}{\lambda_{k+1}} + 16\alpha + 2(1 - \beta)\mathrm{D}_{\mathrm{TV}}(P_d, P_g), \tag{3}$$

*where* $\alpha = \mathbb{E}_{\bar{x} \sim \mathcal{P}_d, x \sim \mathcal{A}(\cdot|\bar{x})} \mathbb{1}[y(x) \neq y(\bar{x})]$ *denotes the labeling error caused by data augmentation,* $\lambda_{k+1}$ *denotes the* $k + 1$-*th smallest eigenvalue of the inflated Laplacian matrix* $\mathcal{L}$, $\mathrm{D}_{\mathrm{TV}}(P_d, P_g) = 1/2 \cdot \int_x |P_d(x) - P_g(x)| dx$ *denotes the total variation (TV) between real and generated data.*[1]

Based on the generalization upper bound in Eq. 3 (proof in Appendix D.2), we can provide rigorous explanations for the AdaInf strategy proposed in Section 3.

### (1) Explaining the Data Inflation Strategy (Section 3.1)

First of all, by involving the generated data, Theorem 4.1 has an additional error term that accounts for the distribution gap between the real and the generated data $\mathrm{D}_{\mathrm{TV}}(P_g, P_d)$, which naturally explains why utilizing a better generative model (with lower FID) brings consistently better downstream performance (Figure 2(a)). Similarly, a larger weight of raw data $\beta$ also helps close the distribution gap, which aligns well with our analysis (Figure 2(b)). [2] Thus, the distribution gap term rigorously justifies the cause from the data inflation side (Section 3.1). For ease of analysis, below we assume that two distributions are roughly the same, *i.e.,* $P_d \approx P_g$.

### (2) Explaining the Data Augmentation Strategy (Section 3.2)

**Influence on Labeling Error** $\alpha$. Second, one would wonder how data inflation affects the labelling error $\alpha$, which, intuitively, means the probability that augmentations produce samples belonging to different classes. As shown in Figure 4(a), stronger augmentations often lead to a larger labeling error. Because $\alpha$ is calculated as the expectation, inflating the data size does not affect it.

**Influence on (Algebraic) Graph Connectivity.** The most critical and interesting part of this analysis, is that we find that data inflation plays an important role in affecting the graph connectivity $\lambda_{k+1}$. From spectral graph theory, we know that Laplacian eigenvalues can serve as algebraic measures of graph connectivity. Loosely speaking, larger eigenvalues indicate better connectivity (and complete graphs have the largest). Data augmentation plays a positive role in improving graph connectivity, since stronger augmentations create more overlap between different training samples (illustrated in Figure 4(b)). Meanwhile, we also notice that using only raw data (*i.e.,* a subset of inflated data) generally has worse connectivity, as there are generally fewer edges when restricted to the subgraph. The following lemma from Chung and Horn (2007) shows that a subsampled graph has a smaller

---

[1]In practice, although we only have finite samples, with the inductive bias of neural networks (Saunshi et al., 2022; HaoChen and Ma, 2023), we will not meet $\lambda_{k+1} = 0$ that renders the bound vacuous.

[2]In practice, since we only use limited real data and training epochs, too large mixing ratio $\beta$ will render the generated data almost useless during training, which also hurt model performance with reduced data diversity. Thus, the optimal reweighting is usually smaller than $0.5$ but not $0$.

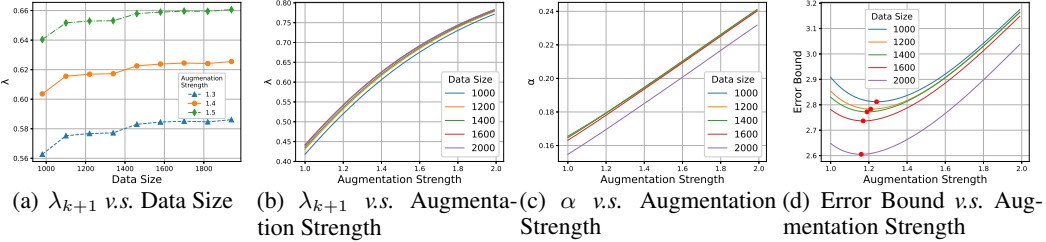

(a) $\lambda_{k+1}$ *v.s.* Data Size    (b) $\lambda_{k+1}$ *v.s.* Augmenta-(c) $\alpha$ *v.s.* Augmentation (d) Error Bound *v.s.* Aug-
tion Strength        Strength       mentation Strength

Figure 5: Analysis of the influence of data size (*i.e.,* inflation) and augmentation strength ($r$) on two crucial factors in the generalization error, the labeling error $\alpha$ and the graph connectivity $\lambda_{k+1}$, on the synthetic dataset (Section 4.3). The optimal augmentation strengths are marked in red dots.

spectral gap (usually equals to the second smallest eigenvalue $\lambda_2$, known as algebraic graph connectivity (Chung, 1997)). Experiments in Appendix B show that other $\lambda_k$'s also decrease with a larger sampling ratio. Since the no-inflation graph can be seen as a subgraph of the inflation graph, it means that inflation will increase the eigenvalues of the non-inflation graph and thus bring better graph connectivity than the raw data.

**Lemma 4.2** (Theorem 1.1 in Chung and Horn (2007)). *Suppose $G$ is a graph on $n$ vertices with spectral gap $\lambda = \min\{\lambda_2, 2 - \lambda_N\}$ and minimum degree $d_{\min}$. A random subgraph $H$ of $G$ with edge-selection probability $p$ almost surely has a spectral gap $\lambda_H$ satisfying*

$$\lambda_H = \lambda - \mathcal{O}\left(\sqrt{\frac{\log n}{p d_{\min}}} + \frac{(\log n)^{3/2}}{p d_{\min}(\log \log n)^{3/2}}\right).$$

**The Complementary Roles between Inflation and Augmentation.** Given the analysis above, we know two important facts: 1) data augmentation has a conflicting effect on downstream performance, since stronger augmentation improves graph connectivity (larger $\lambda_{k+1}$) but also improves labeling error (larger $\alpha$); 2) data inflation only has a one-way effect, as it improves graph connectivity and does not change labeling error. Therefore, when data inflation can bring enough graph connectivity, in order to further minimize the generalization error, we can accordingly adopt a weaker augmentation in the sake of smaller labeling error. In turn, if the data size is too small, we need to adopt stronger augmentation to gain better connectivity. Therefore, *inflation and augmentation have complementary roles for generalization, increasing one of them will decrease the need for the other, and vice versa.* As a result, with more inflated data, the optimal augmentation strength will shift to a lower one, which explains why weak augmentations lead to better performance in Section 3.2.

## 4.3 VERIFICATION EXPERIMENTS

In Section 4.2, we theoretically characterize the influence of data inflation and data augmentation on the generalization error through two crucial factors: labeling error $\alpha$ and connectivity $\lambda_{k+1}$. Since the augmentation graph is hard to construct for real-world data, we now validate this analysis with a synthetic experiment designed following Wang et al. (2022b).

**Setting.** We sample data from the isotropic Gaussian distribution with means $(-1, 0)$ and $(1, 0)$ (two classes), and variance 0.7. The augmentation here is to apply a uniform noise in a circle of radius $r$. Thus, $r$ can be seen as a measure of augmentation strength. With this toy model, we can construct the augmentation graph and explicitly compute the labeling error $\alpha$ and the Laplacian eigenvalues like $\lambda_{k+1}$ ($k = 2$ by default), allowing us to closely examine their changes.

**Results.** Figure 5 shows the influence of data size and augmentation strength, which verifies our analysis from the following aspects. First, Figures 5(a) & 5(b) show that large data size and stronger augmentations are indeed complementary since they both bring better connectivity (larger $\lambda_{k+1}$). Second, Figure 5(c) shows that stronger augmentations indeed bring larger labeling error $\alpha$. At last, Figure 5(d) verifies that when combined, the optimal augmentation (marked by red dots) indeed decreases when the increase of data size, which aligns well with our observation in Section 3.2.

Table 1: Comparison linear probing accuracy (mean & stdev) of different contrastive learning methods and different Datasets. As for the generative models, we adopt STF for CIFAR-10 (1.94 FID) and CIFAR-100 (3.14 FID), and DDPM for Tiny ImageNet (18.61 FID).

(a) Different CL Methods

| Inflation | SimCLR | MoCo V2 | BYOL | Barlow Twins |
|---|---|---|---|---|
| No | 91.56±0.29 | 92.75±0.43 | 92.46±0.06 | 91.24±0.30 |
| Vanilla | 91.38±0.11 | 92.51±0.40 | **92.9**±0.21 | 92.09±0.10 |
| **AdaInf** | **93.42±0.20** | **94.19±0.19** | 92.87±0.26 | **93.64±0.38** |

(b) Different Datasets (w/ SimCLR)

| Inflation | CIFAR-10 | CIFAR-100 | Tiny ImageNet |
|---|---|---|---|
| No | 91.56±0.29 | 66.81±0.36 | 47.21±0.86 |
| Vanilla | 91.38±0.11 | 65.52±0.73 | 41.03±0.39 |
| **AdaInf** | **93.42±0.20** | **69.6±0.21** | **48.36±0.46** |

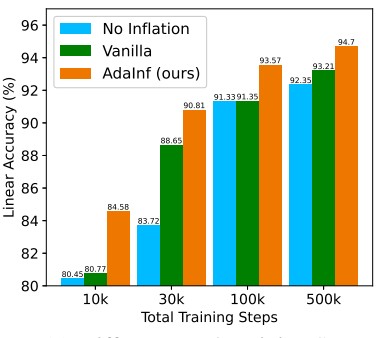

(a) Different Total Training Steps

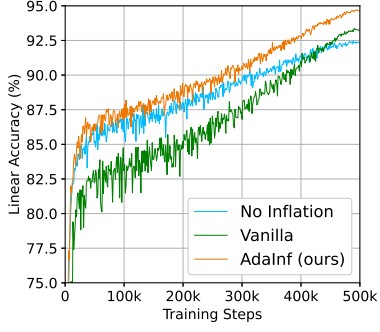

(b) Training Process (500k steps in total)

Figure 6: CIFAR-10 results. (a): linear accuracy (%) with total different training steps (10k steps ≈ 100 epochs of no-inflation training). (b): linear accuracy along the 500k-step training process.

## 5 EXPERIMENTS

**Setup.** We conduct experiments on three benchmark datasets: CIFAR-10, CIFAR-100, and Tiny ImageNet . By default, we use 1M synthetic data for CIFAR-10 and CIFAR-100 generated by a high-quality diffusion model STF (Xu et al., 2023) (FIDs are 1.94 (CIFAR-10) and 3.14 (CIFAR-100)). Due to the limit of computation resource, we adopt DDPM (18.61 FID) for Tiny ImageNet. These diffusion models are unconditional since we only assume access to unlabeled data for pretraining. We evaluate the pre-trained contrastive models using the linear probing method, where only the raw dataset is used. The model training relies on the solo-learn repository (da Costa et al., 2022). We include SimCLR (Chen et al., 2020a) (default choice), MoCo V2 (Chen et al., 2020b), BYOL (Grill et al., 2020), and Barlow Twins (Zbontar et al., 2021) in this part. For a fair comparison of inflated and non-inflated training, we train the model for 100k steps in all cases, which amounts to 1,000 training epochs without inflation. We compare three inflation methods: 1) **No Inflation**, which serves as a baseline for our study; 2) **Vanilla Inflation**, which equally mixes real and generated data and adopts default augmentations; 3) our **AdaInf** strategy, which adopts a mixing ratio of 10 : 1 and weaker augmentations. Specifically, we weaken two most important augmentations: the min scale of random resized cropping improves from 0.08 to 0.2; the ColorJitter strength decreases from 1 to 0.5; and the probability of applying ColorJitter decreases from 0.8 to 0.4.

**Results.** We summarize the benchmark results in Table 1, where we run each experiment for 3 random trials. Table 1(a) shows that compared to the no-inflation baseline, vanilla inflation sometimes leads to even worse performance (*e.g.,* on MoCo V2), while AdaInf has consistent improvements on all datasets. Meanwhile, AdaInf outperforms vanilla inflation significantly (with more than 1% gain in accuracy) on most methods (SimCLR, MoCo V2, and Barlow Twins), while the two perform comparably on BYOL (potentially because the BYOL augmentation needs specific weakening strategy). Table 1(b) shows that AdaInf brings consistent improvements across datasets having different scales and different numbers of classes.

To gain further understanding of data inflation, we further take a closer look at its behaviors. Experiments in this part are conducted on CIFAR-10 with SimCLR unless specified.

Table 2: (a): The impact of weak augmentation and reweighting real data when data inflation. Both of them contribute together to make performance of CL model better. (b): By randomly sampling 5000 images from CIFAR-10 as the original dataset and applying AdaInf for data inflation, the linear accuracy significantly improves.

(a) Ablation Study of AdaInf

| Generated Data | ✗ | ✓ | ✓ | ✓ | ✓ |
|---|---|---|---|---|---|
| Data Reweighting | ✗ | ✗ | ✓ | ✗ | ✓ |
| Weak Augmentation | ✗ | ✗ | ✗ | ✓ | ✓ |
| Linear Accuracy | 91.33 | 91.35 | 91.92 | 93.21 | **93.57** |

(b) Data-Scarce Scenario

| Inflation | Linear Accuracy |
|---|---|
| No Inflation | 74.83 |
| Vanilla | 76.95 |
| AdaInf (ours) | **79.15** |

**Training Steps.** We examine whether data inflation is also effective under different total training steps. As shown in Figure 6(a), AdaInf brings a very large improvement under short training ($80.45 \rightarrow 84.58$ with 10k training steps), and there still remains a clear advantage even if we train for 500k steps (around 5,000 epochs in standard training). Remarkably, SimCLR attains 94.7% linear accuracy under this setting, setting a new SSL record on CIFAR-10 with simply the SimCLR method. It shows that even the simplest method has the potential to match state-of-the-art performance by simply inflating the dataset with generated data, and combining data inflation with advanced methods may lead to further improvements.

**Learning Curve.** We further examine the learning process with and without data inflation in Figure 6(b). Interestingly, we observe that vanilla inflation is inferior to non-inflated training in most time and only achieves a small improvement at last (when non-inflated training saturates). In comparison, AdaInf can consistently outperform the standard non-inflated training across the entire training process, and continue to bring improvements when non-inflated training saturates.

**Ablation Study.** We study the influence of the three components of AdaInf in Table 2(a): generated data, data reweighting, weak augmentation. We can observe that while all three components contribute to the final performance, their importance is: weak augmentation > data reweighting > generated data. In particular, the major improvement is brought by weak augmentation that solely brings around +2% accuracy. It shows that the interplay between inflated data and the learning algorithm has a large influence on the final performance and the adaptive co-design is very important.

**Application to Data-scarce Scenarios.** As generative models can provide a large amount of synthetic data, the proposed data inflation strategy can be particularly helpful when facing data scarcity issues. To show this benefit, we construct a small dataset consisting of 5,000 images (1/10 size of CIFAR-10) by randomly sampling 500 images from each class of CIFAR-10. Subsequently, we train an STF model on this subset (with 18.27 FID) and use the generated 1M samples for data inflation. As shown in Table 2(b), AdaInf obtains much higher linear accuracy than standard training (+4.32% accuracy), and it also achieves better performance than vanilla inflation (+2.2%). Thus, AdaInf is indeed a simple and effective approach for data-scarce scenarios.

## 6 CONCLUSION

In this work, contrary to the common belief that generated data help representation learning, we show that they can be harmful when improperly used for contrastive learning. Upon our investigation, we have identified two sources of failure from the data inflation and data augmentation perspectives. To gain a better understanding of these phenomena, we have provided rigorous theoretical explanations with a careful examination of the generalization bounds. Based on these observations, we propose an adaptive data inflation strategy, Adaptive Inflation (AdaInf), that combines data reweighting and weak augmentations for inflated contrastive learning. Experiments show that this simple data-centric strategy brings significant improvements over the standard training and the vanilla inflation method without any additional cost, especially on data-scarce scenarios.

ACKNOWLEDGEMENT

Yisen Wang was supported by National Key R&D Program of China (2022ZD0160300), National Natural Science Foundation of China (62376010, 92370129), Beijing Nova Program (20230484344), and CCF-BaiChuan-Ebtech Foundation Model Fund.

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

## A    ADDITIONAL RELATED WORK

**Learning with Generated Data.** Using synthetic data from generative models has been explored in many scenarios (Sankaranarayanan et al., 2018; Dosovitskiy et al., 2015; Ma et al., 2022; Guo et al., 2022; Baranchuk et al., 2022). To address the distribution shift between generated data and real data, many works have focused on sampling methods of generative models (Besnier et al., 2020; Wang et al., 2021b; He et al., 2023). Jurjo et al. (2021) proposed that the color, image coherence, and FID of the generative model are important factors influencing the learning representation from generated data. Moreover, Ren and Lee (2018) proposed learning representations by predicting information (*e.g.,* surface normal, depth, and instance contour) of generated data through self-supervised learning. Recently, due to the rise of diffusion models that are able to synthesize high-quality images, generated data have also been extensively studied for enhancing representation learning. Jahanian et al. (2022) and Tian et al. (2023) show that using generated data alone can achieve comparable performance to real data for contrastive learning with proper configurations of the generative models. While a major drawback of these methods is that they often require problem-specific designs of the sampling process which can be costly. Instead, we explore whether generated data with a standard sampling process can help contrastive learning by enlarging the dataset. Azizi et al. (2023) recently show that this kind of generated data can significantly improve supervised learning by around 1% accuracy on ImageNet. In view of these successes, the goal of this work is to deeply investigate how generated data influence contrastive learning, and provide theoretical explanations for these effects.

## B    ADDITIONAL RESULTS

**Influence of Subsampling on Laplacian Eigenvalues.** We randomly generate 10k datapoints from a two-dimensional uniform distribution $[0, 1]^2$ as overall dataset and sample the dataset with different sample ratios. We then construct the augmentation sub-graph by drawing edged $A_{ij} = 1$ for any data pair satisfying $\|x_i - x_j\|_2 \leq 0.05$. Figure 7 shows the Laplacian eigenvalues of the subsampled graphs. We can observe a consistent trend that a smaller sampling ratio leads to a larger decrease of eigenvalues. Therefore, equivalently speaking, using stronger data inflation (*i.e.,* increasing from a smaller sampling ratio (the original data) to 1 (inflated data)) can lead to a larger increase of eigenvalues, and thus bring larger improvements in graph connectivity.

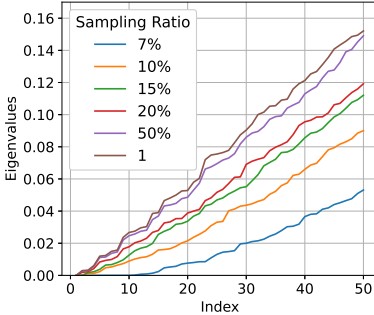

Figure 7: The Laplacian eigenvalues $\lambda_1 \sim \lambda_{50}$ under different sampling ratios.

**Influence of the Scale of Generated Data.** Figure 8 illustrates the impact of different scales of generated data on the linear accuracy of SimCLR. Consistent with the results shown in Figure 2(b), the comparison of results between "No Replication" and "10:1 Replication" in Figure 8 further highlights the significance of replication (Data Reweighting) in mitigating the distribution shift between real data and generated data. Additionally, the results indicate that using 1M generated data samples is the optimal scale when employing the AdaInf strategy. Notably, using a larger size of generated data beyond 1M when 10:1 replication actually leads to worse performance. This is because the same replication leads to a smaller mixing ratio $\beta$ with a larger size of generated data, leading to a larger distribution gap between the real and generated data, as explained in Section 4.2. Therefore,

if we want to utilize a larger amount of generated data than 1M, we should use a larger replication to maintain the same mixing ratio $\beta$.

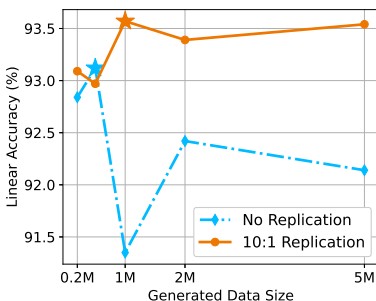

Figure 8: Linear accuracy on inflated dataset with different scale generated data.

**Augmentation Matters.** Figure 3(a) investigates the optimal minimal cropped area size for different-scale datasets. Based on Figure 3(a), Table 3 investigates the effects of both the minimal cropped size and color jitter which play a crucial role in contrastive learning when inflating the original dataset. The results demonstrate that the standard augmentation (the first row) is suitable for no inflation dataset (baseline). However, the weak augmentation method proves to be more effective compared to the standard augmentation. Specifically, when the crop minimum scale is set to 0.2 and the color jitter probability is 0.4 with strength of 0.5 (relative), SimCLR achieves the best performance on inflated data. Once again, these results highlight the complementary effects between data inflation and data augmentation.

Table 3: Linear accuracy (%) of SimCLR with different augmentation strength.

| RRC Min Scale | Color Jitter | | Dataset | |
| | Prob | Strength | CIFAR-10 (Baseline) | Inflated CIFAR-10 |
| --- | --- | --- | --- | --- |
| 0.08 | 0.8 | 1x | **91.33** | 91.92 |
| 0.2 | 0.8 | 1x | 90.81 | 92.98 |
| 0.3 | 0.8 | 1x | 88.27 | 92.68 |
| 0.2 | 0.8 | 0.5x | 91.01 | 93.43 |
| 0.2 | 0.4 | 1x | 90.89 | 93.11 |
| 0.2 | 0.4 | 0.5x | 90.96 | **93.57** |

**The Optimal Mixing Ratio Depends on the Quality of Generated Data.** Figure 9 shows the impact of data quality on the optimal mixing ratio. For generated data from STF ($FID = 1.94$), the optimal replication is $10 : 1$ which means a real sample is roughly worth 10 generated samples under weak augmentation. However, for lower quality generated data from DDPM (FID=3.04), the optimal replication is $15 : 1$ which means a real sample is roughly worth 15 generated samples. The result is consistent with Theorem 4.1, which suggests adjusting the mixing ratio $\beta$ based on the magnitude of the distribution gap.

**Performance on High-Resolution Images.** To further validate the relationship between data scale and data augmentation on high-resolution images, we compared the optimal augmentation strength between two different scale datasets: ImageNet100 (1300 images for each class) and 10%*ImageNet100 (randomly sampled 10% of from each class of ImageNet100, 130 images for each class). The Table 4 shows that for 10%*ImageNet100, the model performs better with a RRC min scale of 0.04 compared to 0.08. However, for ImageNet100, which scale is larger than 10% ImageNet100, weaker augmentation (RRC min scale of 0.08) achieved better performance.

**The Influence of Augmentation under Different Training Steps.** Figure 10 shows the influence of augmentation under different training steps. We can see that under different training steps, using

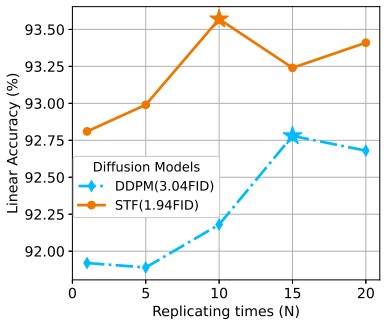

Figure 9: Linear accuracy on different inflated datasets under different data reweighting.

Table 4: Linear accuracy with different augmentation strengths on ImageNet100 (1300 images for each class) and 10% ImageNet100 (randomly sampled 10% of from each class of ImageNet100, 130 images for each class), by changing the min scale of random resized cropping (low value represents stronger augmentation).

| RRC min scale | 10% ImageNet100 | ImageNet100 |
|---|---|---|
| 0.04 | **47.24** | 71.26 |
| 0.08 | 45.34 | **72.76** |

weak augmentation after data inflation is effective. Additionally, we can see that when training is insufficient (fewer steps), generated data tend to help than hurt the performance. Instead, as training converges (100k steps ≈ 1000 epochs), "Inf & Standard Augmentation" underperforms no inflation but our AdaInf still performs well.

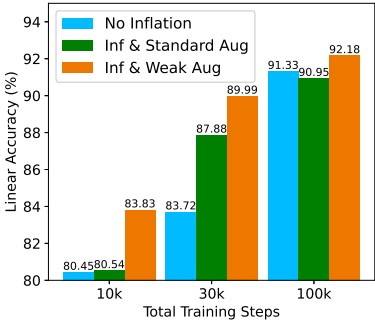

Figure 10: The influence of augmentation under different trainging steps. The experimental configuration for "Inf & Standard Aug" involves applying standard augmentation to the inflated CIFAR-10. The same applies to "Inf & weak augmentation". The generated data is from DDPM (FID=3.04).

**Comparison between GAN and Diffusion Model.** There are mainly two types of generated models with SOTA performance on CIFAR-10: diffusion model and GANs currently. Table 5 provides a comparison of data inflation on CIFAR-10 with generated data from StyleGAN2-ADA (GAN) (Karras et al., 2020) and STF (diffusion model). We find that similar to DDPM, vanilla inflation leads to worse performance, while AdaInf can bring significant improvements. Comparably, diffusion models with better generation quality (e.g., STF with lower FID) can achieve better accuracy.

**Pretraining Cost of the Generative Model.** We conduct a time test for pretraining diffusion models with 4 NVIDIA GTX 3090 GPUs on CIFAR-10, and the total training time is shown in Table 6. We

Table 5: Linear accuracy (%) of SimCLR with generated data from GAN and diffusion model.

| Model | FID | No Inflation | Vanilla Inflation | AdaInf(ours) |
|---|---|---|---|---|
| StyleGAN2-ADA | 2.50 | 91.33 | 90.06 | **91.93 (+1.87)** |
| STF (diffusion) | 1.94 | 91.33 | 91.35 | **93.57 (+2.02)** |

can see that models with better quality (EDM, STF) generally require longer training. In practice, since these models have public checkpoints, we do not need to train on our own for CIFAR-10.

Table 6: Pretraining cost of generative models. All of the models are pretrained with 4 NVIDIA GTX 3090 GPUs.

| Model | DDPM (3.04FID) | STF (1.94FID) | EDM (1.96FID) |
|---|---|---|---|
| Training Time | 71h | 87h | 91h |

## C  EXPERIMENTAL DETAILS

Unless otherwise specified, all experiments are conducted using the default configuration in the codebase of the solo-learn library (da Costa et al., 2022). The CIFAR-10 dataset served as the original dataset and 1M generated data is generated by STF (1.94 FID). We adopt SimCLR with backone Resnet-18 for contrasive learning and pre-train for 100,000 steps. In AdaInf, weak augmentation is used with a minimal cropped (relative) area size of 0.2 and a color jitter probability of 0.4. The values for brightness, contrast, and saturation of color jitter is 0.4 and the hue value is 0.1. Other hyperparameters of weak augmentation remained consistent with the standard augmentation provided by the solo-learn library.

For the experiment in Figure 2(a), the dataset consisted of CIFAR-10 and 1M generated data. The generated data from STF was generated using the provided checkpoint from Xu et al. (2023). The generated data from EDM (1.96 FID) and DDPM was generated using the provided checkpoint from Karras et al. (2022). Additionally, the generated data for EDM (7.29 FID) was generated using a shorter training time checkpoint, where only 5017k images were trained.

For the experiment in Figure 3(a), the augmentation method modified the minimal cropped (relative) area size based on the standard augmentation. The dataset "Half CIFAR-10" was created by randomly selecting 2,500 images from each class of CIFAR-10. The dataset "CIFAR-10 + 1M" consisted of CIFAR-10 data and 1 million generated data from STF and "CIFAR-10 + 0.1M" consisted of CIFAR-10 and 0.1 million generated data from STF.

For the experiment in Figure 3(b), the dataset for the Vanilla method consisted of CIFAR-10 data and 1 million generated data, with standard augmentation. The dataset for the AdaInf method consisted of CIFAR-10 data replicated 10 times and 1 million generated data, with weak augmentation.

For the experiment in Table 1(b), the generated data for CIFAR-100 is from STF, while the generated data for Tiny-ImageNet is from DDPM. The configuration of SimCLR trained on Tiny-ImageNet followed the settings provided by the solo-learn library for imagenet-100.

For the experiment in Table 2(b), we sampled 5,00 images from each class of CIFAR-10 to create a small-scale dataset. We then generated 100,000 images by STF trained on this small-scale dataset, which is used for data inflation. The training steps is 10,000.

# D  OMITTED PROOF

## D.1  PROOF OF THEOREM 3.1

*Proof.* Since $P_t = \beta P_d + (1 - \beta)P_g$ and $0 \le \beta \le 1$, we have

$$
\begin{aligned}
\mathrm{D_{TV}}&(P_t, P_d) \\
&= \int |P_t(x) - P_d(x)| dx \\
&= \int |\beta P_d + (1 - \beta)P_g - P_d(x)| dx \\
&= (1 - \beta) \int |P_g - P_d(x)| dx \\
&= (1 - \beta) \mathrm{D_{TV}}(P_g, P_d),
\end{aligned}
\tag{4}
$$

which completes the proof. $\square$

## D.2  PROOF OF THEOREM 4.1

*Proof.* We define a linear function with weights $B \in \mathbb{R}^{k \times r}$ ($r$ represents the number of dataset classes) as $g_{f,B} : \mathbb{R}^k \to \mathcal{Y}$ on top of pretrained features $f(x) \in \mathbb{R}^k$ to predict the labels $y \in \mathcal{Y}$ of augmentated data. And we define

$$
\bar{g}_{f,B}(\bar{x}) := \arg\max_{i \in [r]} \Pr_{x \sim \mathcal{A}(\cdot|\bar{x})} (g_{f,B}(x) = i)
\tag{5}
$$

to predict the labels $y \in \mathcal{Y}$ of real data.

To describe the difference between labels of two augmented data of the same natural datapoint, we define a function:

$$
\phi^y := \sum_{x,x' \in \mathcal{X}} A_{xx'} \cdot \mathbb{1}[y(x) \ne y(x')].
\tag{6}
$$

We have

$$
\begin{aligned}
\phi^y &= \sum_{x,x' \in \mathcal{X}} A_{xx'} \cdot \mathbb{1}[y(x) \ne y(x')] \\
&= \mathbb{E}_{\bar{x} \sim P_t} \sum_{x,x' \in \mathcal{X}} [\mathcal{A}(x|\bar{x})\mathcal{A}(x'|\bar{x}) \cdot \mathbb{1}[y(x) \ne y(x')] \\
&\le \mathbb{E}_{\bar{x} \sim P_t} \sum_{x,x' \in \mathcal{X}} [\mathcal{A}(x|\bar{x})\mathcal{A}(x'|\bar{x}) \cdot (\mathbb{1}[y(x) \ne y(\bar{x})] + \mathbb{1}[y(x') \ne y(\bar{x})]) \\
&= 2 \cdot \mathbb{E}_{\bar{x} \sim P_t} \sum_{x \in \mathcal{X}} [\mathcal{A}(x|\bar{x}) \cdot \mathbb{1}[y(x) \ne y(\bar{x})]] \\
&= 2 \cdot \mathbb{E}_{\bar{x} \sim P_t} \mathbb{E}_{x \sim \mathcal{A}(\cdot|\bar{x})} \mathbb{1}[y(x) \ne y(\bar{x})] \\
&= 2\alpha
\end{aligned}
\tag{7}
$$

.

**Lemma D.1** (Theorem B.3 in HaoChen et al. (2021)). *Assume the set of augmented data $\mathcal{X}$ is finite. Let $f^* \in \arg\min_{f:\mathcal{X} \to \mathbb{R}^k}$ be a minimizer of the population spectral contrastive loss $\mathcal{L}(f)$ with $k \in \mathcal{Z}^+$. Then, for any labeling function $\hat{y} : \mathcal{X} \leftarrow [r]$ there exists a linear probe $B^* \in \mathbb{R}^{r \times k}$ with norm $\|B^*\|_F \le 1/(1 - \lambda_k)$ such that*

$$
\mathbb{E}_{\bar{x} \sim \mathcal{P}, x \sim \mathcal{A}(\cdot|\bar{x})}[\|\vec{y}(\bar{x}) - B^* f^*(x)\|_2^2] \le \frac{\phi^y}{\lambda_{k+1}} + 4\Delta(y, \hat{y}),
\tag{8}
$$

*where $\vec{y}(\bar{x})$ is the one-hot embedding of $y(\bar{x})$ and $\Delta(y, \hat{y}) := \Pr_{\bar{x} \sim P_t, x \sim \mathcal{A}(\cdot|\bar{x})} (y(x) \ne \hat{y}(x))$ denotes the average disagreement between $\hat{y}$ and the ground-truth labeling $y$.*
*Furthermore, the error can be bounded by*

$$
\Pr_{\bar{x} \sim P_t, x \sim \mathcal{A}(\cdot|\bar{x})} (g_{f^*, B^*}(x) \ne y(\bar{x})) \le \frac{2\phi^y}{\lambda_{k+1}} + 8\alpha.
\tag{9}
$$

According to the definition of $\bar{g}_f(\bar{x})$, $\bar{g}_{f^*,B^*}(\bar{x}) \neq y(\bar{x})$ happens only if more than half of the augmentations of $\bar{x}$ predicts differently from $y(\bar{x})$. Formally, for any $\bar{x} \sim P_t$ with $\bar{g}_{f^*,B^*}(\bar{x})) \neq y(\bar{x})$, according to the definition of $\bar{g}_{f^*,B^*}(\bar{x})$, we must have

$$\Pr_{x \sim \mathcal{A}(\cdot|\bar{x})} (g_{f^*,B^*}(x)) \neq y(\bar{x})) \geq 0.5.$$

Thus we have

$$\Pr_{\bar{x} \sim P_t} (\bar{g}_{f^*,B^*}(\bar{x}) \neq y(\bar{x})) \leq 2 \cdot \Pr_{\bar{x} \sim P_t, x \sim \mathcal{A}(\cdot|\bar{x})} (g_{f^*,B^*}(x)) \neq y(\bar{x})) \leq \frac{8\alpha}{\lambda_{k+1}} + 16\alpha. \tag{10}$$

At last, we notice that the following result is obtained when the training and test data are from the same distribution. When considering generalization from the training distribution $P_t$ to the testing distribution $P_d$, we have

$$\Pr_{\bar{x} \sim P_d} (\bar{g}_{f^*,B^*}(\bar{x})) \neq y(\bar{x}))$$

$$= \int P_d(x) \mathbb{1}[\bar{g}_{f^*,B^*}(\bar{x})) \neq y(\bar{x})] d\bar{x}$$

$$= \int (P_d(x) - P_t(x) + P_t(x)) \mathbb{1}[\bar{g}_{f^*,B^*}(\bar{x})) \neq y(\bar{x})] d\bar{x}$$

$$\leq \int |P_d(x) - P_t(x)| \mathbb{1}[\bar{g}_{f^*,B^*}(\bar{x})) \neq y(\bar{x})] d\bar{x} + \int P_t(x) \mathbb{1}[g_{f^*,B^*}(\bar{x})) \neq y(\bar{x})] d\bar{x} \tag{11}$$

$$\leq \int |P_d(x) - P_t(x)| d\bar{x} + \int P_t(x) \mathbb{1}[\bar{g}_{f^*,B^*}(\bar{x})) \neq y(\bar{x})] d\bar{x}$$

$$= 2 \operatorname{D_{TV}}(P_d, P_t) + \Pr_{\bar{x} \sim P_t} (\bar{g}_{f^*,B^*}(\bar{x})) \neq y(\bar{x}))$$

$$\leq 2(1 - \beta) \operatorname{D_{TV}}(P_d, P_g) + \frac{8\alpha}{\lambda_{k+1}} + 16\alpha.$$

In the last equation, we combine the result of Theorem 3.1 and the generalization bound on training data (Eq. 10). $\qquad\square$

# E EXAMPLES OF GENERATED DATA

For a concrete understanding, we provide examples of the generated data with different diffusion models on different datasets below. It can be seen that the generated data used in our experiments indeed look very similar to the real data, and models with lower FID indeed have fewer artifacts.

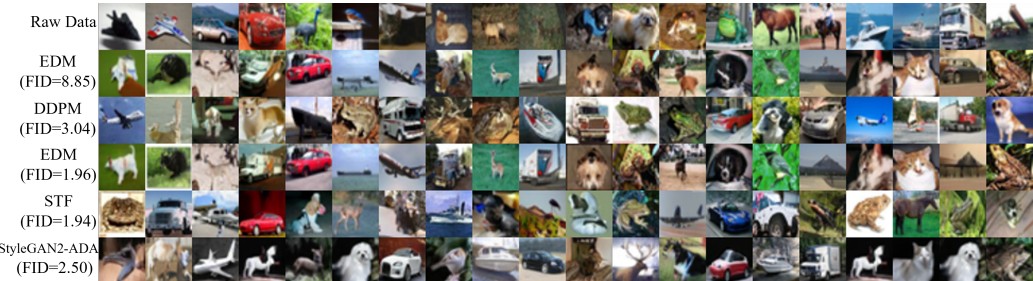

Figure 11: CIFAR-10 examples.

Raw Data

STF
(FID=3.14)

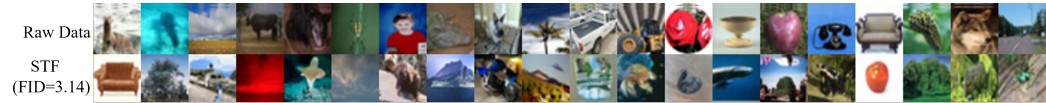

Figure 12: CIFAR-100 examples.

Raw Data

DDPM
(FID=18.61)

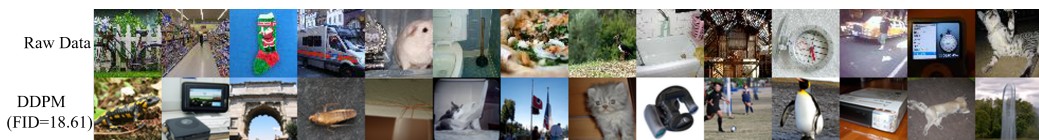

Figure 13: Tiny ImageNet examples.

