# OpenReview forum: "Do Generated Data Always Help Contrastive Learning?"
_ICLR.cc/2024/Conference — ICLR 2024 poster_

### Official Review · Reviewer_XrzN · 2023-10-29

**Soundness:** 3 good
**Presentation:** 3 good
**Contribution:** 3 good
**Rating:** 6
**Confidence:** 4

**Summary:**

This paper explored the reason for the performance degradation when utilizing synthetic data from a generative model to enhance contrastive learning (CL). The authors found that the strengths of the data augmentation and the mixing ratio or the size of data inflation are the two critical factors that affect performance through theoretical analysis and empirical results. Based on the observation, the author proposed AdaInf, which adapts the weight of synthetic data and the strengths of the data augmentation to achieve the best performance. Experimental results on multiple datasets showed that AdaInf can outperform the plain CL method and the CL method with naive data inflation.

**Strengths:**

The strengths of the paper are listed as follows.
1. This paper is well-motivated. Data inflation is an important method to improve the performance of CL. The reason for the unexpected performance degradation is worth exploring.
2. The conclusions about the data inflation and data augmentation are analyzed theoretically and proved with a simple synthetic dataset empirically.
3. The paper is well-written, where the problem is clearly defined and the analysis is easy to follow.

**Weaknesses:**

The weaknesses of the paper are listed as follows.
1. It would be better if the authors could compare AdaInf with the other existing data inflation methods for CL.
2. The description of AdaInf is in qualitative way. For example, it claims to use different weights on real and generated data and adopt milder data augmentation. However, how to get an appropriate weighted and the strengths of the data augmentation remains unknown.
3. It would be better if the authors could provide some experiments on more generative models like GAN.

**Questions:**

The questions of the paper are listed as follows.
1. For equation (1), should there be a negative sign before the first term?
2. Are there any experiments to show that the two tricks in AdaInf can also improve the performance of CL with data generated by more generative models like GANs?
3. On page 3, there is a statement that better generative quality often requires larger models and/or slower sampling. What does the slower sampling mean?
4. Are there any experiments to show the trade-off between the size of generated data for inflation and the strength of augmentation on real-world datasets? It looks like all the experiments used the same size of generated data.
5. The purpose of Figure 4 is a little bit confusing to me. Why can this figure be used to show stronger augmentations create more overlap between different training samples?
6. For the experiments of Figure 6, is the batch size for CL, CL with vanilla data augmentation and CL with AdaInf the same? If this is the case, does it mean the training cost is the same when the total training steps are the same for the three methods?
7. What is the pertaining cost of the diffusion model on the evaluated datasets?

---

> ### Author Response · Authors · 2023-11-18
> **Response to Reviewer XrzN (1/2)**
>
> Thank you for your detailed reading and for acknowledging the novelty and theory of our work. We address your main concerns below.
>
> ---
>
> **Q1**. It would be better if the authors could compare AdaInf with the other existing data inflation methods for CL.
>
> **A1**. Upon our survey of the literature, there are several works studied **only using generative models** for SL / CL [1,2,3], and one explored combining real and generated data for supervised learning [4]. As far as we see, **no existing work explores the mixed training strategy for self-supervised CL as done in ours**. Another key difference is that they often utilize generative models trained from **external datasets or supervision**, which leads to an unfair comparison. Instead, we make sure that only **the unsupervised pretraining dataset** is available for all settings, and we utilize the same dataset to train the generative model and perform contrastive learning.
>
> **References:**
>
> [1] Yonglong Tian, Lijie Fan, Phillip Isola, Huiwen Chang, and Dilip Krishnan. Stablerep: Synthetic images from text-to-image models make strong visual representation learners. arXiv preprint arXiv:2306.00984, 2023.
>
> [2] Ali Jahanian, Xavier Puig, Yonglong Tian, and Phillip Isola. Generative models as a data source for multiview representation learning. In ICLR, 2022.
> [3] Victor Besnier, Himalaya Jain, Andrei Bursuc, Matthieu Cord, and Patrick Pérez. This dataset does not exist: Training models from generated images. In ICASSP, 2020.
> [4] Ruifei He, Shuyang Sun, Xin Yu, Chuhui Xue, Wenqing Zhang, Philip H. S. Torr, Song Bai, and Xiaojuan Qi. Is synthetic data from generative models ready for image recognition? In ICLR, 2023.
>
> ---
>
> **Q2**. The description of AdaInf is in qualitative way. For example, it claims to use different weights on real and generated data and adopt milder data augmentation. However, how to get an appropriate weighted and the strengths of the data augmentation remains unknown.
>
> **A2**. Since our theoretical and empirical discoveries are general and not limited to a particular reweighting or augmentation strategy, we use **AdaInf to refer to a general designing principle for data inflation, that we should adapt data reweighting and data augmentation to the chosen diffusion model and inflated data size.** The word “adaptive” here is used to highlight the data dependence.
>
> Nevertheless, since modern CL methods adopt multiple hand-designed augmentations — each with their own hyperparameters (eg cropping, color jittering) — it is hard to find a universal law for adjusting them, and it is a common practice to select a well-performing default configuration (e.g., SimCLR, MoCo, BYOL). Thus, we also resort to a default strategy that works well across multiple datasets. **The proposed AdaInf principle can provide guidelines for large-scale pretraining with generated data, and our simple AdaInf strategy can serve as a good starting point that delivers significant gains.** With large enough computing, one may further uncover a scaling law for AdaInf, and we leave it for future work. We have revised the definitions of AdaInf in  Sec 3.3 to avoid confusion.
>
> ---
>
> **Q3**. It would be better if the authors could provide some experiments on more generative models like GAN.
>
> **A3**. Sure we can. Following your suggestion, we have also added evaluations on GANs as another SOTA generative model, also with good generation quality. We find that similar to DDPM, vanilla inflation leads to worse performance, while AdaInf can bring significant improvements. Comparably, diffusion models with better generation quality (e.g., STF with lower FID) can achieve better accuracy. The results are added to **Appendix A**.
>
> *Inflated contrastive learning with 1M generated images on CIFAR-10.*
>
> | Model | FID | No Inflation | Vanilla Inflation | AdaInf |
> | --- | --- | --- | --- | --- |
> | StyleGAN2-ADA | 2.50 | 91.33 | 90.06 | 91.93 (+1.87%) |
> | STF (diffusion) | 1.94 | 91.33 | 91.35 | 93.57 (+2.02%) |
>
> ---
>
> **Q4**. For equation (1), should there be a negative sign before the first term?
>
> **A4**. Yes. We have fixed it in the revision.
>
> ---
>
> **Q5**. Are there any experiments to show that the two tricks in AdaInf can also improve the performance of CL with data generated by more generative models like GANs?
>
> **A5**. Please see **A3**.
>
> ---
>
> **Q6**. On page 3, there is a statement that better generative quality often requires larger models and/or slower sampling. What does the slower sampling mean?
>
> **A6**. To obtain better generation quality in the diffusion model, one way is to adopt many intermediate denoising steps when drawing a sample (e.g., 1000 steps in DDPM), which slows the sampling speed significantly. We have added a short explanation here in the revision.

---

> ### Author Response · Authors · 2023-11-18
> **Response to Reviewer XrzN (2/2)**
>
> **Q7**. Are there any experiments to show the trade-off between the size of generated data for inflation and the strength of augmentation on real-world datasets? It looks like all the experiments used the same size of generated data.
>
> **A7**.  We note that in Appendix A, we discussed the influence of generated data size on linear accuracy (Figure 8), where 1M data generally provide the best performance with data reweighting. Following your suggestion, we further investigate the influence of generated data size on the optimal augmentation strength. As shown in the revised Figure 3a (no data reweighting), 0.1M generated data attain optimal performance with min crop 0.2, while 1M generated data attain the optimal performance with min crop 0.3. **It aligns with the general trend in Fig 3a, showing that a larger data size attains optimal performance with a weaker augmentation**.
>
> ---
>
> **Q8**. The purpose of Figure 4 is a little bit confusing to me. Why can this figure be used to show stronger augmentations create more overlap between different training samples?
>
> **A8**. We are sorry for the confusion. We have revised this figure to be clearer. Now Figure 4b shows that weak augmentations still preserve the main image semantics, so different images cannot be effectively bridged together. Instead, **strong augmentations can create overlap views between different samples through common objects**, such as the wheels in the two images. In this way, stronger augmentations create more overlap between different training samples.
>
> ---
>
> **Q9**. For the experiments of Figure 6, is the batch size for CL, CL with vanilla data augmentation and CL with AdaInf the same? If this is the case, does it mean the training cost is the same when the total training steps are the same for the three methods?
>
> **A9**. For a fair analysis, we adopt the same training configuration (including batch size) for all experiments: CL, CL with vanilla data augmentation, and CL with AdaInf. Thus, as you said, the training cost is indeed the same when the total training steps are the same for the three methods.
>
> ---
>
> **Q10**. What is the pretraining cost of the diffusion model on the evaluated datasets?
>
> **A10**. We conducted a time test for pretraining diffusion models with 4 NVIDIA GTX 3090 GPUs on CIFAR-10, and the total training time is shown below. We can see that diffusion models with better quality (EDM, STF) generally require longer training. In practice, since these models have public checkpoints, we do not need to train on our own for CIFAR-10.
>
> | Model | DDPM | STF | EDM |
> | --- | --- | --- | --- |
> | Training Time | 71h | 87h | 91h |
>
> ---
>
> Thank you again for your careful reading. We hope that the explanations above could address your concerns. We are very happy to address your remaining concerns during the discussion stage.

---

> > ### Comment · Reviewer_XrzN · 2023-11-21
> > **Q7-Q10 are Well Addressed**
> >
> > Thanks for the detailed explanations the authors provided to my questions. I have read the explanations and believe that Q7-Q10 are well addressed.

---

> ### Comment · Reviewer_XrzN · 2023-11-21
> **Q1-Q6 are Almost Addressed**
>
> Thanks for the detailed feedback from the reviewers. Most of my questions are addressed. However, for question 1, I think they already exist on the CL with generative models pretraining without an extra dataset. I will list some of them as follows.
>
> [1]. Jahanian, Ali, et al. "Generative Models as a Data Source for Multiview Representation Learning." International Conference on Learning Representations. 2021.
> [2]. Wu, Yawen, et al. "Synthetic data can also teach: Synthesizing effective data for unsupervised visual representation learning." Proceedings of the AAAI Conference on Artificial Intelligence. 2023.
>
> Could you please compare your work with this type of work including the comparison on performance? I think such kind of comparison is helpful to find the position of your work in the area.

---

> ### Author Response · Authors · 2023-11-22
> **Further Response to Reviewer XrzN**
>
> Thank you for appreciating our response! We will address your remaining concern below.
>
> ---
>
> **Q11**. However, for question 1, I think they already exist on the CL with generative models pretraining without an extra dataset. I will list some of them as follows.
>
> > [1]. Jahanian, Ali, et al. "Generative Models as a Data Source for Multiview Representation Learning." International Conference on Learning Representations. 2021.
> >
>
> > [2]. Wu, Yawen, et al. "Synthetic data can also teach: Synthesizing effective data for unsupervised visual representation learning." Proceedings of the AAAI Conference on Artificial Intelligence. 2023.
> >
>
> Could you please compare your work with this type of work including the comparison on performance? I think such kind of comparison is helpful to find the position of your work in the area.
>
> **A11**. Thank you for reminding us. We will explain the relationships between these two papers and our work as follows.
>
> **Paper [1].** We note that although both exploring the use of generated data, [1] explores a rather different setting to ours: they run CL **with only generated data (trying to compete with real data)**, while we study how to properly **mix the real and the generated data**. In other words, **they focus on how to create better generated data, while we study how to properly train CL with a given set of generated data.** Thus, we believe that the two works are rather orthogonal and can potentially benefit from each other. Since the two explore different settings as stated above, it would be unfair to directly compare their performance.
>
> **Paper [2].** In comparison, [2] mixes real and generated data to CL, which is indeed similar to our setting. The major differences are:
>
> - our method focuses on how to **leverage samples directly sampled from an existing diffusion model,** where similar to [1], they explore how to alter the generative process to get better samples
> - our AdaInf is purely data-centric and model-agnostic (thus bring no computation overhead), and their requires a joint training procedure, which could be heavy and tricky.
>
> Unfortunately, we find that there is **no public implementation of [2],** and the joint training procedure is sophisticated to implement. So, we are afraid that we cannot reproduce this experiment within the limit of time. But we will definitely reach the authors for the code, and if available, we will add a comparison in the updated version.
>
> For now, we can compare some of the reported results in [2] with ours (Table 1) for a rough glimpse:
>
> | Method | Backbone | CIFAR-10 | CIFAR-100 |
> | --- | --- | --- | --- |
> | SimCLR (base) [2] | ResNet-50 | 90.37 | 63.93 |
> | SimCLR + proposed method [2] | ResNet-50 | 92.94 | 67.41 |
> | SimCLR (base) (ours) | ResNet-18 | 91.56 | 66.81 |
> | SimCLR + AdaInf (ours) | ResNet-18 | **93.42** (highest **94.7** with 500k steps) | **69.6** |
>
> In comparison, we report higher scores for the SimCLR baseline, and **achieve higher linear accuracy with a smaller model (ResNet-18 vs ResNet-50)**. Further noting that our method is much simpler and more efficient than theirs, we believe that this comparison shows a clear advantage of the AdaInf method. Since they are rather orthogonal strategies, combining AdaInf and [2] may lead to further improvements.
>
> ---
>
> Thank you again and hope the clarification above could address your concerns. Please let us know if there is more to clarify.

---

### Official Review · Reviewer_umDF · 2023-11-01

**Soundness:** 3 good
**Presentation:** 4 excellent
**Contribution:** 3 good
**Rating:** 6
**Confidence:** 4

**Summary:**

This paper studies the problem of using of synthetic data from generative models (e.g. diffusion models) for contrastive learning (CL). Firstly it finds that naively inflating CIFAR-10 with 1M generated images can in fact hurt linear probe performance of CL. The paper delves deeper in the role of (a) weight of inflated data, (b) augmentation strength. Interestingly the paper finds two simple fixes, (a) upweighting real data by 10x, (b) using weaker augmentations with more data. Based on this finding, the paper proposes adaptive Inflation (AdaInf) strategy that adjusts augmentation strength based on dataset size. Experiments on CIFAR and TinyImageNet evaluate AdaInf data strategy with SimCLR, MoCo V2, BYOL, Barlow Twins. Overall AdaInf does better than no data inflation and data inflation w/ standard augmentation. It achieves accuracy as high as 94.7% on CIFAR-10 with SimCLR. Ablations on smaller training horizon and smaller training data further shows the benefit of AdaInf.

The paper provides theoretical analysis using augmentation graph framework introduced in HaoChen et al. and shows a complementary benefit of inflation and augmentation. Stronger augmentation improve "graph connectivity" at the cost of "labeling error". More data also improves graph connectivity, thus we can get away with weaker augmentation that has smaller labeling error. Synthetic example with gaussian distribution verifies this insight.

**Strengths:**

Significance: The paper studies an interesting problem of role of synthetic data from generative models on contrastive learning. Given the increasing of procuring high quality data, effectively using synthetic data is a important problem to study

Novelty: The perspective on interplay between larger data and augmentation strength is interesting and novel to my knowledge. The value of real data being more than synthetic data is intuitive but not surprising. The theory, although a simple application of HaoChen et al., does provide interesting insights.

Clarity: The paper is overall well written and easy to follow. Experimental results are also presented well.

**Weaknesses:**

- It would have been useful to have some more discussion on the issues of using synthetic data, e.g. bias

- Some clarification and cleanup of the theory parts would be useful. See comments below

Overall the paper makes a positive contribution in studying the interplay between synthetic data and augmentation for contrastive learning. I have currently assigned weak accept, but I'd be happy to raise the score after clarification on the theory.

**Questions:**

- Role of reweighting: Theorem 3.1 and 4.1 do not fully explain that there’s an optimal weighting. In light of mismatch between $P_s$ and $P_d$ Theorem 3.1 basically suggests that \beta=1 is best. I believe the argument in Theorem 4.1 is that $\lambda_{k+1}$ is larger when $P_g$ has larger support. It would be nice to make this point more clear. Overall it would help to more carefully deal with the case of $P_s \neq P_g$ and why would one pick $\beta < 1$.

- Lemma 4.2 uses random sub-graph with probability $p$ of selecting each edge. Is random edge selection reflective of subsampling data? Random node selection seems more appropriate.

- Thm 4.1 argues about connectivity and non-zero $\lambda_{k+1}$ for the training set, but Saunshi et al. 2022 show lack of connectivity in the training set for standard augmentations, implying $\lambda_{k+1}=0$. This would make the bound in Theorem 4.1 vacuous without any . Does that contradict the claims? The results in HaoChen et al. are non-vacuous because that uses the eigenvalue of the population distribution in the bound rather than eigenvalue of training data.


Saunshi et al. 2022. Understanding Contrastive Learning Requires Incorporating Inductive Biases

---

> ### Author Response · Authors · 2023-11-18
> **Response to Reviewer umDF**
>
> Thanks for your detailed reading and for appreciating the discovered interplay between generated data and data augmentation. We are happy to address the theoretical concerns that you mentioned below.
>
> ---
>
> **Q1**. It would have been useful to have some more discussion on the issues of using synthetic data, e.g. bias.
>
> **A1**. Indeed, using generated data will have downsides on generalization, particularly if the generated data is biased, e.g., a distribution gap $D(P_g,P_d)$. **Our Theorem 4.1 theoretically characterizes the influence of this distribution gap on downstream generalization, and shows that data reweighting can help mitigate this bias.** For concerns on social bias (if this is what you were referring to), we believe that generated data reflect the intrinsic bias in real data (if their distributions are close enough). Please let us know if there is more to clarify.
>
> ---
>
> **Q2**. Role of reweighting: Theorem 3.1 and 4.1 do not fully explain that there’s an optimal weighting. In light of mismatch between $P_s$ and $P_d$ Theorem 3.1 basically suggests that $\beta=1$ is best. I believe the argument in Theorem 4.1 is that $\lambda_{k+1}$ is larger when $P_g$ has larger support. It would be nice to make this point more clear. Overall it would help to more carefully deal with the case of $P_s\neq P_g$ and why would one pick $\beta<1$.
>
> **A2**. This is a very good question! We note that the original motivation of using generated data is to enhance the diversity of training data, and choosing $\beta=1$ (no inflation) contradicts this goal. In practice, since we only use *limited training epochs*, too large mixing ratio $\beta\approx$ $1$ will render the generated data almost useless during training, which also hurts model performance with reduced data diversity. Thus, the optimal reweighting is usually close to $1$ but not $1$. Since our generalization bound considers the global optimal solution (instead of finite training), it does not fully reflect this issue. We have added a footnote on page 6 to elaborate on this part.
>
> ---
>
> **Q3.** Lemma 4.2 uses random sub-graph with probability $p$ of selecting each edge. Is random edge selection reflective of subsampling data? Random node selection seems more appropriate.
>
> **A3**. We note that in the InfoNCE loss (Eq. 1), the positive samples are drawn as a pair from the data, which represents an edge in the graph. In other words, during contrastive learning, the influence of inflation size is reflected in the form of sampled edges. Thus, we believe that it also makes sense to consider edge subsampling for modeling their difference. Another important reason is that upon our research, we only found an existing result of Chung and Horn (2007) guaranteeing the spectral gap on edge subsampling. If you happen to know any node subsampling results, we would be happy to include them as well.
>
> ---
>
> **Q4.** Thm 4.1 argues about connectivity and non-zero $\lambda_{k+1}$ for the training set, but Saunshi et al. 2022 show lack of connectivity in the training set for standard augmentations, implying $\lambda_{k+1}=0$**.** This would make the bound in Theorem 4.1 vacuous without any . Does that contradict the claims? The results in HaoChen et al. are non-vacuous because that uses the eigenvalue of the population distribution in the bound rather than eigenvalue of training data.
>
> **A4**. This is also a very good question! Indeed, [1] argued that finite data may suffer from zero augmentation overlap and thus $\lambda_{k+1}=0$. If it is really true, contrastive learning should have bad generalization (proven in [2]). However, in practice, it is not. [1] attributed this to the inductive bias of neural networks that implicitly bring augmentation overlap, and [3] established a formal guarantee on it. Therefore, with the help of architectural inductive bias, we will not have zero augmentation overlap even with finite training data. Since incorporating inductive bias is rather independent of the main interest of this work, we omit this part for simplicity. For completeness, we have added a footnote to explain it on page 6.
>
>  **References:**
>
> [1] Saunshi et al. Understanding Contrastive Learning Requires Incorporating Inductive Biases. In ICML. 2022.
>
> [2] Wang et al. Chaos is a ladder: A new theoretical understanding of contrastive learning via augmentation overlap. In ICLR, 2022.
>
> [3] Haochen and Ma. A Theoretical Study of Inductive Biases in Contrastive Learning. In ICLR. 2023.
>
> ---
>
> Thank you again for your careful reading. We hope that the explanations and revisions above could address your concerns. We are very happy to address your remaining concerns during the discussion stage.

---

### Official Review · Reviewer_pGvU · 2023-11-01

**Soundness:** 2 fair
**Presentation:** 2 fair
**Contribution:** 2 fair
**Rating:** 5
**Confidence:** 3

**Summary:**

This paper discovered the failure mode of data inflation, a technique that augments real data with synthetic data, for contrastive learning. The finding was that data reweighting and weak augmentation can help improve the performance of contrastive learning on the combined dataset. Based on that, they proposed a method, called Adaptive Inflation (AdaInf), to train contrastive learning by adaptively adjusting augmentation strength and mixing ratio for data inflation. In addition, the authors provided the first theoretical guarantees for inflated contrastive learning. Empirically, AdaInf improved downstream accuracy significantly.

**Strengths:**

- This paper is well-structured.
- The theoretical results and findings are interesting.
- The experimental design is good. Experiments and ablation studies successfully verify theoretical results.

**Weaknesses:**

- The preliminary section is not well-written.
- There is a concern about the generality of AdaInf. It seems that the method is hand-crafted rather than adaptive.
- There is a lack of motivation for considering diffusion models in this paper. All theoretical results can be generalized to any generative model.
- Some notation and experiment results are not consistent.

**Questions:**

- Eq. (1) is wrong. In addition, the authors should cite this paper for Eq (1): “Understanding Contrastive Representation Learning through Alignment and Uniformity on the Hypersphere”.
- Eq. (2) is also wrong. Again, the citations are not chosen well. Ho et. al 2020 should be a better reference here.
- Why do we need to sample 1M data while CIFAR-10 has only 60k samples? Speaking of which, instead of increasing $\beta$ we can decrease $|\mathcal{D}_g|$.
- Section 3.1: Is the ratio $10:1$ true for all datasets? And is it universal for different diffusion models?
- Section 3.2: Are there any intuitions or reasonings why larger data size needs weaker data augmentation? Or do generated images need weaker data augmentation and they dominate the inflated data set?
- Section 3.3: How to choose the correct strength of data augmentation in AdaInf? The word "adaptive" seems misleading here because the hyperparameters are chosen manually rather than auto adjusted or learned.
- Section 5.1: Why is the min scale of random resized cropping (0.2) different from the optimal value (0.3) in Figure 3?
- The result in Table 3 is slightly different from that in Figure 3 where 0.3 yields the best performance.
- Figure 8: Is the replication N fixed to 10? Again, the question is whether N = 10 is universal.
- The labeling error definition seems weird. It is better to define it as Eq (8). Then, the term $\phi^y$ in Eq (6)  is less than $2 \alpha$. If the definition is kept as in Theorem 4.1, we have $\phi^y = \alpha$. Note that $\alpha$ in HaoChen et al. 2021a has a different meaning.

**Minors**:
- The usage of $B^\star$ can be safely removed.
- There are a lot of typos: Diffusion models in Section 2, Theorem 4.1, Lemma 4.2, Eq (7), Eq (9), and Eq (12) just to name a few.
- Notation:
  - N is used for replication times in Section 3.1 but later used in Section 4.1 with different meanings. What is the meaning of N in Section 4.1?
  - Eq (6), what is $\mathcal{X}$?
  - Is $w_{x x'}$ defined as $A_{x, x'}$ in the main text?
  - Overall, the notation is similar to HaoChen et al. 2021a. However, the way the authors use them is not good and sufficient.

---

> ### Author Response · Authors · 2023-11-18
> **Response to Reviewer pGvU (1/3)**
>
> Thanks for your detailed reading and appreciating the novelty and soundness of our work. We have revised the paper carefully following your suggestions and fixed the typos that you mentioned. Below, we address your main concerns on the paper content.
>
> ---
>
> **Q1.** The preliminary section is not well-written.
>
> **A1**. Thanks for your careful reading. We have carefully revised the writing and fixed the typos that you mentioned (more details below). Please let us know if there is more to modify.
>
> ---
>
> **Q2**. There is a concern about the generality of AdaInf. It seems that the method is hand-crafted rather than adaptive.
>
> **A2**. Since our theoretical and empirical discoveries are general and not limited to a particular reweighting or augmentation strategy, we use **AdaInf to refer to a general designing principle for data inflation, that we should adapt data reweighting and data augmentation to the chosen diffusion model and inflated data size.** The word “adaptive” here is used to highlight the data dependence.
>
> Nevertheless, since modern CL methods adopt multiple hand-designed augmentations — each with their own hyperparameters (e.g., cropping, color jittering) — it is hard to find a universal law for adjusting them, and it is a common practice to select a well-performing default configuration (e.g., SimCLR, MoCo, BYOL). Thus, we also resort to a default strategy that works well across multiple datasets. **The proposed AdaInf principle can provide guidelines for large-scale pretraining with generated data, and our simple AdaInf strategy can serve as a good starting point that delivers significant gains.** With large enough computing, one may further uncover a scaling law for AdaInf, and we leave it for future work. We have revised the definitions of AdaInf in Sec 3.3 to avoid confusion.
>
> We also acknowledge that the name Adaptive Inflation could be a little confusing despite general principles, we only offer a default choice in practice. We would like to know which alternative name you may find suitable.
>
> ---
>
> **Q3**. Some notation and experiment results are not consistent.
>
> **A3**. Thank you for your detailed reading and kind suggestions. We have fixed the typos and writing errors in the revision (details below).
>
> ---
>
> **Q4.** Eq. (1) is wrong. In addition, the authors should cite this paper for Eq (1): “Understanding Contrastive Representation Learning through Alignment and Uniformity on the Hypersphere”.
>
> **A4.** Thanks. We have fixed it.
>
> ---
>
> **Q5.** Eq. (2) is also wrong. Again, the citations are not chosen well. Ho et. al 2020 should be a better reference here.
>
> **A5**. Thanks. We have fixed it.
>
> ---
>
> **Q6.** Why do we need to sample 1M data while CIFAR-10 has only 60k samples? Speaking of which, instead of increasing $\beta$ we can decrease $|\mathcal{D}_g|$.
>
> **A6**. This is because using more generated data can bring diverse input and facilitate generation. We have shown the gain of using more generated data in **Figure 8**. As quoted below, decreasing  $|\mathcal{D}_g|$ from 1M to 0.2M will lead to worse performance (about -0.5%). **Even without reweighting (first row)**, we need to inflate **at least 0.5M** generated samples (10x CIFAR-10) to attain the best performance of ****93.12%. With reweighting, we can allow to use a larger 1M dataset, and **attain a better performance of 93.57%**. Based on this evidence, we adopt 1M dataset + reweighting as our default strategy.
>
> | Generated data　size | 0.2M | 0.5M | 1M | 2M | 5M |
> | --- | --- | --- | --- | --- | --- |
> | No replication | 92.84 | 93.12 | 91.35 | 92.42 | 92.14 |
> | 10:1 Replication | 93.09 | 92.97 | 93.57 | 93.39 | 93.54 |

---

> ### Author Response · Authors · 2023-11-18
> **Response to Reviewer pGvU (2/3)**
>
> **Q7**. Is the ratio $10:1$ true for all datasets? And is it universal for different diffusion models?
>
> **A7**. First, we want to note that 10:1 is indeed a good default choice for diffusion models and datasets. Simply comparing Fig 2a (no reweighting) and Fig 3b (with 10:1 reweighting), we will find that the 10:1 reweighting improve **all diffusion models**. Further, in Tab 1, we show that AdaInf with 10:1 reweighting also brings consistent improvement on different CL methods (Tab 1a) and datasets (Tab 1b). Therefore, **10:1 is indeed a very good default choice** that works for all the settings studied in this work.
>
> Meanwhile, given further computation budget, one may finetune the reweighting to find a better one. For example, since DDPM has worse generation quality than STF, according to our theory, it should have less weight on generated data to avoid large distribution gap. As shown below, indeed a slightly larger reweighting ratio 15:1 works better for DDPM, which aligns well with our analysis.
>
> To summarize, we believe that 10:1 is a good default choice that works across multiple settings, and further tuning it may find a slight better reweighting coefficient.
>
> | Replication | 1 | 5 | 10 | 15 | 20 |
> | --- | --- | --- | --- | --- | --- |
> | DDPM (FID=3.04) | 91.92 | 91.89 | 92.18 | 92.78 | 92.68 |
> | STF (FID=1.94) | 92.81 | 92.99 | 93.57 | 93.24 | 93.41 |
>
> ---
>
> **Q8**. Section 3.2: Are there any intuitions or reasonings why larger data size needs weaker data augmentation? Or do generated images need weaker data augmentation and they dominate the inflated data set?
>
> **A8**. Yes! To understand it more thoroughly, we give a theoretical characterization on this phenomenon in Sec 4.2, and the intuitive logic is as follows. As pointed out by many papers [1,2], contrastive learning generally relies on **strong data augmentation to create connectivity** between samples, in order to get good downstream generalization. However, strong augmentation also has downsides as it **introduces large labeling errors** with distribution shifts. To resolve this issue, **we wish to use as weak augmentations as possible if only we could have the same level of connectivity,** while data inflation is exactly such a technique that can improve graph connectivity by enlarging the training set. Therefore, as pointed out in Sec 4.2, data inflation and data augmentation have complementary effects. With inflation sharing the burden of graph connectivity, we can use weaker augmentation to minimize its downsides (labeling error) and gain better downstream performance.
>
> > Or do generated images need weaker data augmentation and they dominate the inflated data set?
> >
>
> This seems a common misunderstanding of our method. Although it is an intuitive logic, it is **not exactly true**. Fig 3a shows that **subsampling real data (Half CIFAR-10) would also require stronger data augmentation (min crop 0.02) to attain the optimal performance.** In turn, it means that scaling from Half CIFAR-10 and CIFAR-10 **with real data** also requires weaker augmentation (min crop 0.08). Therefore, it is NOT the real-synthetic distribution gap that requires us to adjust data augmentation**,** but the inflated data size. ****Indeed, ****our theory above suggests that even if the synthetic data are perfect ($P_d=P_g$), it is also helpful to decrease data augmentation strength under more data inflation, which aligns with the observation in Fig 3a.
>
> **References:**
>
> [1] Jeff Z HaoChen, Colin Wei, Adrien Gaidon, and Tengyu Ma. Provable guarantees for self-supervised deep learning with spectral contrastive loss. In NeurIPS, 2021.
>
> [2] Yifei Wang, Qi Zhang, Yisen Wang, Jiansheng Yang, and Zhouchen Lin. Chaos is a ladder: A new theoretical understanding of contrastive learning via augmentation overlap. In ICLR, 2022.
>
> ---
>
> **Q9**. Section 3.3: How to choose the correct strength of data augmentation in AdaInf? The word "adaptive" seems misleading here because the hyperparameters are chosen manually rather than auto adjusted or learned.
>
> **A9**. Please see **A2** above.
>
> ---
>
> **Q10.** Section 5.1: Why is the min scale of random resized cropping (0.2) different from the optimal value (0.3) in Figure 3? The result in Table 3 is slightly different from that in Figure 3 where 0.3 yields the best performance.
>
> **A10**. In Figure 3, we do not add data reweighting (1:1) for ablating their effects, while we adopt 10:1 reweighting in Table 3 for a comprehensive analysis, so there is a slight difference. Overall, 0.2 and 0.3 perform similarly (gap $\leq$ 0.3%) in two settings.
>
> ---
>
> **Q11**. Figure 8: Is the replication N fixed to 10? Again, the question is whether N = 10 is universal.
>
> **A11**. Yes, we adopt 10:1 across all experiments in the experiment section and appendix section. For the latter question, please see **A7**.

---

> ### Author Response · Authors · 2023-11-18
> **Response to Reviewer pGvU (3/3)**
>
> **Q12**. The labeling error definition seems weird. It is better to define it as Eq (8). Then, the term $\phi^y$ in Eq (6) is less than $2\alpha$. If the definition is kept as in Theorem 4.1, we have $\phi^y=\alpha$. Note that $\alpha$ in HaoChen et al. 2021a has a different meaning.
>
> **A12**. Thanks for pointing it out! Indeed, we made a typo here, and it would be better to define $\alpha$ as Eq. 8, which is consistent with Haochen et al. We have fixed this in the revision.
>
> ---
>
> **Q13**. The usage of $B^*$ can be safely removed.
>
> **A13**. Here, $B^*$ denotes the optimal weight of the linear classifier. We have revised the formulations in Sec 4 to be clearer.
>
> ---
>
> **Q14**. There are a lot of typos: Diffusion models in Section 2, Theorem 4.1, Lemma 4.2, Eq (7), Eq (9), and Eq (12) just to name a few.
>
> **A14**. Thanks for pointing them out! We have fixed these typos in the revision.
>
> ---
>
> **Q15**. Notations.
>
> Thanks for pointing it out! We have revised them in the revision.
>
> > N is used for replication times in Section 3.1 but later used in Section 4.1 with different meanings. What is the meaning of N in Section 4.1?
> >
>
> Yes. We abuse the notations here. We change $N$ in Sec 4.1 to $n$, where it denotes the number of training data.
>
> > Eq (6), what is $\mathcal{X}$?
> >
>
> It denotes the set of all augmented data. We have added definitions.
>
> > Is $w_{xx'}$ defined as $A_{xx'}$ in the main text?
> >
>
> Yes. We have modified them to be $A_{xx'}$ to be consistent.
>
> ---
>
> Thank you again for your careful reading. We have carefully refined our paper and fixed all the typos following your suggestions, and addressed each of your concerns above. We respectfully suggest that you could re-evaluate our work based on the updated results. We are very happy to address your remaining concerns on our work.

---

> > ### Comment · Reviewer_pGvU · 2023-11-22
> > **Response to authors' rebuttal**
> >
> > I thank the authors for patiently reading and responding to all my questions. I found that my concern was adequately addressed. In addition, the revised paper has improved in terms of both the related works and methodology parts. However, I still find the diffusion models have a loose connection with the theoretical results in Sections 3 and 4.
> >
> > All in all, I would like to increase my score to 5.

---

> > > ### Author Response · Authors · 2023-11-23
> > > **Further Response to Reviewer  pGvU**
> > >
> > > Thank you for appreciating our response and for increasing the score! We are happy to address your remaining concerns.
> > >
> > > ---
> > >
> > > **Q16.** I still find the diffusion models have a loose connection with the theoretical results in Sections 3 and 4.
> > >
> > > **A16.** We want to note that the data inflation strategy explored in this work is **model-agnostic**, and **we can use *any generative models* that have good generation quality, in theory, a lower distribution gap $D_{\rm TV}(P_d,P_g)$.** We pick diffusion models only because it has SOTA generation quality to date. As shown in the revision **(Appendix A)**, our AdaInf also **delivers significant improvements over vanilla inflation with StyleGAN2 (quoted below)**. This model-agnostic property makes AdaInf more generally applicable than strategies specifically designed for one class of models.
> > >
> > > Accordingly, our theory is also general and not specified for diffusion models. Thus, the theory also fits for our practice as well.
> > >
> > > *Inflated contrastive learning with 1M generated images on CIFAR-10.*
> > >
> > > | Model | FID | No Inflation | Vanilla Inflation | AdaInf |
> > > | --- | --- | --- | --- | --- |
> > > | StyleGAN2-ADA | 2.50 | 91.33 | 90.06 | **91.93 (+1.87%)** |
> > > | STF (diffusion) | 1.94 | 91.33 | 91.35 | **93.57 (+2.02%)** |
> > >
> > > ---
> > >
> > > Hope this explanation could address your concerns! Please let us if there is more to clarify.

---

> > > > ### Comment · Reviewer_pGvU · 2023-11-23
> > > > **Response**
> > > >
> > > > That is exactly what I mentioned earlier in the weaknesses part: "There is a lack of motivation for considering diffusion models in this paper. All theoretical results can be generalized to any generative model." My concern is more about the writing/presentation of the paper itself.

---

> > > > > ### Author Response · Authors · 2023-11-23
> > > > >
> > > > > Thanks for your explanation. We now get your point.  We will revise the writing to present the results in a more general sense.

---

> > > > > > ### Author Response · Authors · 2023-11-23
> > > > > >
> > > > > > Deare Reviewer  pGvU,
> > > > > >
> > > > > > We have revised our paper to present the results in a broad context. Specifically, we remove some expressions that are specified to diffusion models, and add a general discussion on generative models in the background section. We will continue to polish the writing to be more accurate, and include the GAN result in the main paper once we obtain more complete results. Thanks for your suggestion.
> > > > > >
> > > > > > In the above discussions, we have addressed your concerns in detail, and we are delighted to hear that you find them "adequately addressed". We thus respectfully suggest that you could re-evaluate our work with the updated results and revisions.
> > > > > >
> > > > > > Authors

---

### Official Review · Reviewer_17xG · 2023-11-06

**Soundness:** 2 fair
**Presentation:** 3 good
**Contribution:** 2 fair
**Rating:** 3
**Confidence:** 3

**Summary:**

The paper studies whether synthetic data generated with a diffusion model can help self-supervised learning. By reweighting the ratio of generated and real data, and re-tuning the augmentation strength used on the positive pairs, the authors are able to show that gains are possible on CIFAR10, CIFAR100 and Tiny Imagenet.

**Strengths:**

The paper is generally well written, and the results are interesting (albeit limited in scale, see below, which limits signficance). Some theoretical intuition is given, although clarity in the connection of theory and empirical results could be improved.

**Weaknesses:**

Theory and empirical experiments are not well connected, in my opinion (I would be happy to get corrected on this during the rebuttal). Here is how I read the paper: Leveraging synthetic data requires to define how it will be mixed with real data. Also, as image statistics of synthetic data might be different from real data*, the existing augmentations applied to contrastive learning might be suboptiomal --- hence, re-tuning both aspects of the model could help to boost performance. If the authors would have a better way to connect theory and empiricaly observations, this could be interesting --- but in the current outline of the paper, both tracks seem pretty independent to me.

Another weakness are the empirical results: These results are fairly small scale with limited evaluation of different generative models. A lot of evaluations are performed on CIFAR-10 where one would expect the possibility to run extensive experiments and analysis of various factors. To make the statements in the paper more convincing, it would help to extend these experiments, compute error bars, study the influence of other parameters, and extend the most promising models towards full ImageNet-scale.

Additional weaknesses/concerns:

- The InfoNCE loss on (1) is incorrect. It misses a "-" in front of the positive loss, and also typically in InfoNCE, there is a sum vs. a mean over the negative examples, as the number of negatives influences embedding quality (this only yields a $\log N$ difference in the value of the loss but does not influence the gradients) --- I think it is still better to re-state the loss as common in the literature.
- "significantly" is used at different places in the paper. Please check the usage, and remove where it is not backed up by a statistical test. I would strongly encourage the authors to run multiple (I would suggest at least 5) seeds of their models and report error bars on every CIFAR-scale experiment reported in the paper.
- in Table 1, the FID scores of the used generative models on the respective dataset should be added, this seems to be a major influence on experiment outcome.

Minor

- Theorem 3.1, was the total variation introduced before? If not, clarify the abbreviation (TV) where it first appears in the text.
- for the theorems, it might be useful to link to the respective section where the proof is stated, for easier navigation through the document
- The notation in 4.1 should be double checked. E.g. $\mathcal{\overline{X}} = {\overline{x}}$, is that correct? or $D_{xx}$.
- Sec 4.2, "an downstream" -> "a downstream"

**Questions:**

- could you add some samples from the diffusion models to the supplement for a better visual impression at different FID levels? ideally compared to a set of samples from the real datasets.
- do you use different diffusion models (with different training datasets) for each of the considered subtasks? in particular, is e.g. the CIFAR model only trained on CIFAR, etc?
- Sec 4.2, how is $P_d \approx P_g$ a reasonable assumption in the light of your argument for stating Theorem 3.1?
- How do you connect Lemma 4.2 to your observed empirical results, what is it signifiance?
- In Fig 6., Vanilla data inflation *always* helps or keeps results the same. How does this relate to the statements in Figure 3(b)? It seems like the number of training steps is an important confounder we need to control for?
- datasets like ImageNet have a "photographer"/object centric bias. Do the generated samples also have such a bias? If not, could this be a possible explanation for the difference in the optimal augmentation strategy?
- Is the augmentation strategy in the base setup optimal with respect to the selection metric? If so, how was this confirmed?

---

> ### Author Response · Authors · 2023-11-18
> **Response to Reviewer 17xG (1/4)**
>
> Thank you for your careful reading and pointing out the problems. We have carefully revised our paper following your suggestions. Below we address your main concerns.
>
> ---
>
> **Q1.** Theory and empirical experiments are not well connected, in my opinion (I would be happy to get corrected on this during the rebuttal).
>
> **A1. First**, let us explain how we connect theory and practice. After we discover the two AdaInf strategies in Sec 3, we provide a rigorous theoretical justification in Sec 4. In particular,  Theorem 4.1 establishes **a theoretical bound that takes the reweighting-related terms ($\beta$ and $D_{\rm TV}(P_d,P_g)$) and the augmentation-related terms ($\alpha$ and $\lambda_{k+1}$) into consideration**. We then provide **a detailed explanation in Sec 4.2** to show that the theoretical bounds imply the empirical phenomena that we observed. To make it clearer, we now explicitly link each explanation to each strategy in the revision (see page 6).
>
> **Second**, we want to point out that there are some misunderstandings in your descriptions below.
>
> > Here is how I read the paper: Leveraging synthetic data requires to define how it will be mixed with real data. Also, as image statistics of synthetic data might be different from real data, the existing augmentations applied to contrastive learning might be suboptiomal --- hence, re-tuning both aspects of the model could help to boost performance.
> >
>
> We highlight that in our theoretical analysis, the distribution gap is the cause of the **data reweighting strategy** (because reweighting can provably close the gap)**, but NOT the cause of the data augmentation** strategy. Instead, **even if the synthetic data are perfect ($P_d=P_g$)**, it is also helpful to decrease data augmentation strength under more data inflation. Fig 3a gives the evidence by showing that **subsampled real data (Half CIFAR-10) would also require stronger data augmentation to attain the optimal performance, and vice versa.** We theoretically justify this phenomenon with three paragraphs in Sec 3, and the logic is as follows. Since data inflation enhances the connectivity of augmentation graph, we can achieve a smaller labeling error under the same connectivity, so the optimal augmentation strength will become smaller.
>
> So, to be precise, the takeaway of our theoretical analysis is that, ***for data inflation, we could adjust reweighting coefficients to mitigate the distribution gap, and adjust augmentation strength to fully unleash the benefits of enlarged data size**.* In this way, we can gain more from generated data and benefit downstream tasks a lot.
>
> Therefore, we believe that **our empirical findings can be sufficiently explained by our theoretical analysis in Sec 4.2.** As noted by Reviewer gCqn, ***“the results would greatly help in our understanding of contrastive learning both theoretically and empirically.**”* Hope this explanation could address your concerns. Please let us know if there is more to clarify.

---

> ### Author Response · Authors · 2023-11-18
> **Response to Reviewer 17xG (2/4)**
>
> **Q2.** Another weakness are the empirical results. These results are fairly small scale with limited evaluation of different generative models. To make the statements in the paper more convincing, it would help to extend these experiments, compute error bars, study the influence of other parameters, and extend the most promising models towards full ImageNet-scale.
>
> **A2.** In our work, we mainly explore the effect of **augmenting original data with a lot more generated data (so we call it data inflation)**. For example, we augment CIFAR-10 of 50k samples with 1M generated samples (20x inflation). It is well-known that generating high-quality samples from diffusion models is very computationally expensive. For now, the largest dataset we are able to cope with is Tiny-ImageNet (see Table 1), where training a diffusion model and generating 1M 64x64 samples take us two whole weeks. And with limit of computation, we have tried our best to evaluate the proposed method on diverse datasets (CIFAR-10, CIFAR-100, Tiny ImageNet) **with different data sizes (Tiny Imagenet is of 1/10 ImageNet size), different numbers of classes (10 & 100), and different resolutions (32x32, 64x64)**. Thus, we believe that our experiments can also justify the generality of AdaInf across different kinds of data.
>
> **Error bars.** **We have added the error bars to the reported results following your suggestions.** From Table 1, we can see that AdaInf indeed significantly improves over the vanilla inflation.
>
> **Generative models.**  We have also added evaluations on GANs as another SOTA generative model, also with good generation quality. We find that similar to DDPM, vanilla inflation leads to worse performance, while AdaInf can bring significant improvements. Comparably, diffusion models with better generation quality (e.g., STF with lower FID) can achieve better accuracy. The results are added to **Appendix A**.
>
> *Inflated contrastive learning with 1M generated images on CIFAR-10.*
>
> | Model | FID | No Inflation | Vanilla Inflation | AdaInf |
> | --- | --- | --- | --- | --- |
> | StyleGAN2-ADA | 2.50 | 91.33 | 90.06 | 91.93 (+1.87%) |
> | STF (diffusion) | 1.94 | 91.33 | 91.35 | 93.57 (+2.02%) |
>
> **Large-scale study.** Studying inflation on large-scale data, e.g., inflating ImageNet-1k (1M size) by 20x, would require **huge computing** (generating 1M ImageNet images (each 2s) requires 23 days, and 20M requires 460 days) which is far beyond our capability. Within the limit of rebuttal time, we propose to validate our method on ImageNet by studying two subsets: ImageNet-100 (1300 images for each class) and 10% ImageNet100 (randomly sampled 10% off from each class of ImageNet100, 130 images for each class). According to our analysis, a larger dataset (ImageNet-100) would require a stronger augmentation to attain the optimal performance.
>
> **Results.** As shown below, fewer data (10% ImageNet-100) attain better performance with stronger augmentation (0.04 RRC min scale), while full ImageNet-100 performs better with weaker augmentation (0.08 RRC min scale). It aligns very well with our theoretical and empirical observation on a large-scale and high-resolution experiment: a larger data size (e.g., with inflation) requires a weaker augmentation to attain the optimal performance. Thus, our conclusion also holds for ImageNet-scale datasets.
>
> | RRC min scale | 10% ImageNet100 | ImageNet100 |
> | --- | --- | --- |
> | 0.04 | **47.24** | 71.26 |
> | 0.08 | 45.34 | **72.76** |

---

> ### Author Response · Authors · 2023-11-18
> **Response to Reviewer 17xG (3/4)**
>
> **Q3.** The InfoNCE loss on (1) is incorrect. It misses a "-" in front of the positive loss, and also typically in InfoNCE, there is a sum vs. a mean over the negative examples, as the number of negatives influences embedding quality (this only yields a  difference in the value of the loss but does not influence the gradients) --- I think it is still better to re-state the loss as common in the literature.
>
> **A3.** Thanks for pointing it out! We have revised the writing and adopted the original form.
>
> ---
>
> **Q4.** "significantly" is used at different places in the paper. Please check the usage, and remove where it is not backed up by a statistical test.
>
> **A4.** Due to the limit of time to reproduce all experiments, we run 3 times for each experiment in Table 1. It can be seen that AdaInf can indeed obtain statistically significant improvement over the baseline. We have also revised our writing accordingly following your suggestion.
>
> ---
>
> **Q5.** Table 1, the FID scores of the used generative models on the respective dataset should be added, this seems to be a major influence on experiment outcome.
>
> **A5.** We note that the default diffusion models and their FID scores used in Table 1 are introduced in the Setup paragraph in the beginning of Section 5, as quoted below.
>
> > *By default, we use 1M synthetic data for CIFAR-10 and CIFAR-100 generated by a high-quality diffusion model STF (Xu et al., 2023) (FIDs are 1.94 (CIFAR-10) and 3.14 (CIFAR-100)). Due to the limit of computation resource, we adopt DDPM (18.61 FID) for Tiny ImageNet. These diffusion models are unconditional since we only assume access to unlabeled data for pretraining.*
> >
>
> We have also added it to the caption of Table 1 for better clarity in the revision.
>
> ---
>
> **Q6.** Theorem 3.1, was the total variation introduced before? If not, clarify the abbreviation (TV) where it first appears in the text.
>
> **A6.** Thanks. We have added definitions here.
>
> ---
>
> **Q7.** for the theorems, it might be useful to link to the respective section where the proof is stated, for easier navigation through the document
>
> **A7.** Thanks. We have added links to the proof.
>
> ---
>
> Q8. The notation in 4.1 should be double checked. E.g., $\bar{\mathcal{X}}=\{\bar x\}$ is that correct? or $D_{xx}$.
>
> **A8.** Here, we want to say that $\bar{\mathcal{X}}$ are composed of samples like $\bar x$. For $D_{xx}$, as in [1], $x$ is used as an index, so $D_{x,x}$ is the $(x,x)$-th element of $D$. We add descriptions to avoid confusion.
>
> **Reference:**
>
> [1] Jeff Z HaoChen, Colin Wei, Adrien Gaidon, and Tengyu Ma. Provable guarantees for self-supervised deep learning with spectral contrastive loss. In NeurIPS, 2021.
>
> ---
>
> **Q9.** Sec 4.2, "an downstream" -> "a downstream"
>
> **A9.** Thank you and we have fixed it.
>
> ---
>
> **Q10.** Could you add some samples from the diffusion models to the supplement for a better visual impression at different FID levels? ideally compared to a set of samples from the real datasets.
>
> **A10.** Sure. We have added randomly selected samples from real and generated data (from different diffusion models) in Figures 11, 12, and 13 in Appendix E. It can be seen that the generated data used in our experiments indeed look very similar to the real data, and models with lower FID indeed have fewer artifacts.
>
> ---
>
> **Q11**. Do you use different diffusion models (with different training datasets) for each of the considered subtasks? in particular, is e.g. the CIFAR model only trained on CIFAR, etc?
>
> **A11**. Yes. For a fair comparison, we make sure that **we do not utilize any additional supervision or dataset** (highlighted in Sec 2, p3). To do so, we first train an unsupervised diffusion model on the given dataset, use the dataset to generate synthetic data, and train the model with the mixed data.
>
> ---
>
> **Q12.** Sec 4.2, how is $P_d\approx P_g$ a reasonable assumption in the light of your argument for stating Theorem 3.1?
>
> **A12.** As elaborated in **A1**, we only need to deal with the distribution gap with data reweighting, while adjusting augmentation also benefits downstream generalization a lot even when the generated data are perfect, i.e., $P_d=P_g$. Accordingly, in Sec 4.2, **we want to disentangle the influence of these two parts**, so we assume $P_g=P_d$ when discussing the influence on $\alpha,\lambda$ caused by data augmentation. Empirically, the diffusion models that we utilize (e.g., STF) have very high generation quality (very low FIDs) and as shown in **Appendix E,** the generated samples look very similar to real CIFAR images and Tiny ImageNet images (Figures 11, 12, 13). So it is also a reasonable assumption in practice.

---

> ### Author Response · Authors · 2023-11-18
> **Response to Reviewer 17xG (4/4)**
>
> **Q13**. How do you connect Lemma 4.2 to your observed empirical results, what is its significance?
>
> **A13**. Lemma 4.2 states that a smaller sampling ratio $p$ of a graph leads to a larger decrease of graph connectivity (also intuitively true). In our scenario, assuming perfect generation $P_d=P_g$, the original real data can be seen as the subsampled set of the inflated data, say $D\subseteq \tilde{D}$. Therefore, according to Lemma 4.2, **inflating with more generated data corresponds to a larger increase of connectivity** (exactly the point we want to make in this paragraph in page 6). Further combining this fact with the two-way effect of data augmentation (increasing both labeling error and connectivity), we arrive at the conclusion that **the optimal augmentation strength will become weaker with more inflated dat**a, so as to attain lower downstream generalization error (elaborated in the last paragraph of page 6). **This theoretical result aligns well with our empirical observations in real-world data (Sec 3.2, Fig 3a) as well as synthetic experiments (Sec 4.3).**
>
> ---
>
> **Q14**. In Fig 6., Vanilla data inflation *always* helps or keeps results the same. How does this relate to the statements in Figure 3(b)? It seems like the number of training steps is an important confounder we need to control for?
>
> **A14**. The key point here is that in all experiments in the experiment section (including Fig 6), **we adopt the diffusion model STF by default** to explore the limit of AdaInf. As shown in **Fig 3b**, the quality of the diffusion model matters. As the SOTA model, STF with 1.94 FID can help with standard augmentation (only marginally from 91.33% $\to$ 91.92%). Instead, if we adopt a medium model, like the popular **DDPM with 3.04 FID, we will have worse performance** (90.95% with reweighting and 90.27% with no reweighting (Fig 2a)) than the baseline (91.33%), under standard augmentation.
>
> **Number of training steps.** We further study the influence of training steps following your suggestion. From the table below, we can see that when training is insufficient (fewer steps), generated data tend to help with more diverse data. Instead, **as training converges (100k steps), the standard augmentation underperforms no inflation** with a large distribution gap and suboptimal augmentation. Our AdaInf can contribute to mitigate these two issues by adopting reweighting and weaker augmentation.
>
> *Contrastive learning on CIFAR-10 with **DDPM (FID=3.04)** generated data for different training steps (with 10:1 reweighting, which improves performance on both augmentations (Fig 2b)).*
>
> | Method | Training steps |  |  |
> | --- | --- | --- | --- |
> |  | 10k | 30k | 100k |
> | No Inflation | 80.45 | 83.72 | 91.33 |
> | Vanilla（No reweighting + Standard Augmentation） | 80.38 | 87.36 | 90.27 |
> | Reweighting+Standard Augmentation | 80.54 | 87.88 | 90.95 |
> | AdaInf（Reweighting+Weak Augmentation） | 83.83 | 89.99 | 92.18 |
>
> ---
>
> **Q15.** Datasets like ImageNet have a "photographer"/object centric bias. Do the generated samples also have such a bias? If not, could this be a possible explanation for the difference in the optimal augmentation strategy?
>
> **A15**. Since the generated data are very close to the real data (with low FID), they also have the object centric bias, as shown in the examples in **Figures 11,12,13 in Appendix E.** Since they are the same on this point, we are afraid that it cannot explain the difference in optimal augmentation strategy. As we explained in **A1** and further verified in **A2**, the difference in optimal augmentation is not caused by distribution shift, but by the difference in **data size**.
>
> ---
>
> **Q16**. Is the augmentation strategy in the base setup optimal with respect to the selection metric? If so, how was this confirmed?
>
> **A16**. Indeed, the default augmentation strategy (e.g., RRC with 0.08 min scale) is optimal under the base setup (no inflation). We can see this from **Fig 3a**, where training with original CIFAR-10 indeed peaks at 0.08 RRC scale (the default parameter). Thus, the base setup is indeed optimal.
>
> Instead, training with Half CIFAR-10 data peaks at stronger augmentation (smaller min scale, 0.02), and training with more (inflated) data peaks at weaker augmentation (larger min scale: 0.2 with 0.1M generated data, 0.3 with 1M generated data). This nicely justifies our point that larger data size requires weaker augmentations in contrastive learning.
>
> ---
>
> Thank you again for your careful reading. We have carefully refined our paper according to your suggestions, and address each of your concerns above. We respectfully suggest that you could re-evaluate our work based on these updated results. We are very happy to address your remaining concerns on our work.

---

> ### Author Response · Authors · 2023-11-23
>
> Dear Reviewer 17xG,
>
> We have carefully prepared a detailed response to address each of your questions.
>
> In particular, as you said, *"If the authors would have a better way to connect theory and empiricaly observations, this could be interesting"*. We have elaborated how our theory can well explain the observed empirical phenomena in our response, and have also revised the paper to make them clearer.
>
> Would you please take a look and let us know whether you find it satisfactory?
>
> Thanks! Have a great day!
>
> Authors

---

### Official Review · Reviewer_gCqn · 2023-11-10

**Soundness:** 3 good
**Presentation:** 3 good
**Contribution:** 4 excellent
**Rating:** 8
**Confidence:** 3

**Summary:**

The paper challenges the common belief of generated data being always helpful to contrastive learning and empirically show that this need not always be the case. The paper identifies, two issues that control if generated data would be helpful or not to learn good representations, namely the discrepancy between generated and real data distributions and the mixing ratio of real and generated data. Based on these insights, the authors propose AdaInf which does a grid search over different mixing ratios and augmentation levels to find the best combination for downstream tasks. The authors also give theoretical justifications as to the empirical trends they observe.

**Strengths:**

The insights and results provided by this paper seem novel, creative and significant. I believe the results would greatly help in our understanding of contrastive learning both theoretically and empirically.

The paper is very well-written in general except some proofs in the appendix which I will describe in detail in the weaknesses section.

**Weaknesses:**

The proposed Adainf method seems to rely on the downstream task to find the optimal weighing factor and augmentation strength. This might limit it's use in practice since the goal of self supervised learning is to learn a good representation from training data that can be useful in future downstream tasks, I do not know a priori. Some discussion on this would be useful, either as a limitation or clarifying the exposition in case I misunderstood something.

Eq (7) in proof of theorem 4.1: I think the steps need to be explained. It is not clear to me how the indicator function is swapped from I(y(x) \neq y(x')) to I(y(x) \neq y(\bar{x})) in the second equality, why is the first inequality true, and how the sum disappeared in the third equality.

What is the difference between the definition of \alpha and equation 8? Shouldn't they be equal?

The statement of theorem 4.1 should be qualified with what exactly is the optimal encoder? What is the objective here? Lemma D.1 indicates this is some population spectral contrastive loss. The precise definition of this loss should be made clear in the paper. Similarly, eq 5 should be in the main text since it defines the prediction function using the linear classifier on top of Pretrained features. The paper makes no mention that this is really majority voting over prediction made on all possible augmentations of the sample point.

I would suggest the authors use Markov's inequality to obtain equation 11 to make the proof rigorous.

Minor Typo: Eq (12) second equality, shouldn't it be P_t instead of P_g?

**Questions:**

See weaknesses.

---

> ### Author Response · Authors · 2023-11-18
> **Response to Reviewer gCqn**
>
> Thank you for your careful reading and encouraging comments! We have carefully revised the paper (including the proof part) following your suggestions. We address your additional concerns as follows.
>
> ---
>
> **Q1.** The proposed Adainf method seems to rely on the downstream task to find the optimal weighing factor and augmentation strength. Some discussion on this would be useful, either as a limitation or clarifying the exposition in case I misunderstood something.
>
> **A1.** Indeed, in the main paper, we mainly consider image classification as the downstream task for the theoretical and empirical analysis. But we believe that it does not harm the generality of the proposed approach. We will elaborate this with the following three points.
>
> **First**, this is a common practice for designing SSL methods and choosing hyperparameters, as done in SimCLR, MoCo, etc. Notably, existing practice (e.g., the SimCLR paper) shows that augmentations selected according to classification generally have **good transfer performance** on other downstream tasks (eg detection, segmentation).
>
> **Second**, Table 1b shows that the default AdaInf strategy (designed based on CIFAR-10) also has **good transferability when directly applied to other datasets** like CIFAR-100 and Tiny ImageNet. Therefore, we believe that **even on a new dataset, one can directly apply the default AdaInf strategy and obtain significant improvements**.
>
> **Third**, if we really need to re-tune the AdaInf strategy solely based on a pretraining dataset, we can rely on surrogate evaluation metrics developed in previous work to find an optimal AdaInf strategy. For example, ARC [1] is shown to have good alignment with downstream accuracy (correlation > 90%) using only pretrained data. Since this direction is somewhat orthogonal to our study, we leave it for future work.
>
> Hope this could ease your concern. We have added discussions in the main paper to elaborate this (Sec 3.3). Please let us know if there is more to clarify.
>
> **Reference:**
>
> [1] Wang et al. Chaos is a Ladder: A New Theoretical Understanding of Contrastive Learning via Augmentation Overlap. In ICLR. 2022.
>
> ---
>
> **Q2.** Eq (7) in proof of theorem 4.1: how the indicator function is swapped from I(y(x) \neq y(x')) to I(y(x) \neq y(\bar{x})) in the second equality, why is the first inequality true, and how the sum disappeared in the third equality.
>
> **A2.** We apologize for the confusion. There are some typos here. In the second equality, it should be $1[y(x)\neq y(x')]$ instead of $1[y(x)\neq y(\bar x)]$. Thus, the first inequality is a triangular inequality as we insert $\bar x$. In the third equality, the sum should be still there (but degrades to $x$ alone). We have fixed these typos in the revision.
>
> ---
>
> **Q3.** What is the difference between the definition of \alpha and equation 8? Shouldn't they be equal?
>
> **A3.** Here is another typo. The definition of alpha should be $\mathbb{E}_{\bar{x} \sim P_t,x \sim \mathcal{A}(\cdot \mid \bar{x})}\mathbb{1}[y(x) \neq y(\bar{x})]$. We have checked across the paper and fixed them in the revision.
>
> ---
>
> **Q4.** The statement of theorem 4.1 should be qualified with what exactly is the optimal encoder, and the prediction function.
>
> **A4.** In Theorem 4.1, we omitted some technical details to be more concise and easier to read. We have now added more background in the formulation (Sec 4.1) to be more explicit and rigorous following your suggestion.
>
> ---
>
> **Q5.** I would suggest the authors use Markov's inequality to obtain equation 11 to make the proof rigorous.
>
> **A5.** Thanks for your suggestion! Actually, we justified this inequality with the definition of the average classifier, since a misclassification implies that ${Pr}(g_{f, B}(x)) \neq y(\bar{x})) \geq 0.5$. This classic result can be traced back to Lemma 4.1 in [1], where they use a similar justification like ours. We have added a more detailed derivation in the revision.
>
> **Reference:**
>
> [1] Langford, John, and John Shawe-Taylor. PAC-Bayes & margins. In NeurIPS. 2002.
>
> ---
>
> **Q6.** Eq (12) second equality, shouldn't it be P_t instead of P_g?
>
> **A6.** Yes! We have fixed it in the revision.
>
> ---
>
> Thank you again for your detailed reading and constructive comments, which helped improve this work! Please let us know if there is more to clarify. We are happy to address them during the discussion stage.

---

### Author Response · Authors · 2023-11-18
**Paper Update**

We sincerely thank all reviewers for their detailed reading and valuable comments. We have carefully responded their concerns, and incorporated these suggestions in the updated manuscript of 18 pages. The main revisions are:

- Sec 2: revise preliminary to be more rigorous and readable.
- **Sec 3.1: add tradeoff analysis of the optimal augmentation strength with different generated data sizes (Figure 3a)**
- Sec 3.3: state the Adaptive Inflation principle clearer
- Sec 4.1: revise the theoretical formulation to be clearer
- Sec 4.2: explicitly link theoretical analyses to each phenomenon
- **Sec 5:  report mean and stdev results obtained from multiple runs (Table 1)**
- **Appendix A: add many new analysis results** on 1) the optimal mixing ratio, 2) high-resolution images, 3) the influence of augmentation under different training steps, 4) **new generative models (GAN);** and 5) pretraining cost.
- Appendix D: add more details on the proof to be more rigorous and fix the typos.
- **Appendix F: illustrate random examples generated by different generative models.**

---

### Meta-Review · Area_Chair_ShCq · 2023-12-12

**Metareview:**

This paper presents a new approach to using synthetic data for contrastive learning, introducing the Adaptive Inflation (AdaInf) strategy. The reviewers recognize the significance of this work in addressing the challenges associated with data inflation in the context of contrastive learning. The reviewers also recognize theoretical insights into the interplay between data augmentation strength and data mixing ratios, which contribute to a deeper understanding of the role of synthetic data in contrastive learning. Empirical results across various datasets, including CIFAR and TinyImageNet, have shown the effectiveness of the AdaInf method. However, some concerns were raised regarding the scope of empirical evaluation, particularly the lack of extensive experiments on larger datasets like ImageNet. Additionally, there were also comments on the need for clearer connections between the theoretical aspects and empirical observations, as well as the practical implications of hyperparameter tuning. Despite these concerns, the paper represents a valuable contribution to the field, offering novel perspectives and methodologies that can spur further research and discussion within the community. Therefore, I recommend accepting this paper.

**Justification For Why Not Higher Score:**

Lacking of larger scale experiments to demonstrate the gain.

**Justification For Why Not Lower Score:**

important topics and interesting ideas

---

### Decision · Program_Chairs · 2024-01-16

Accept (poster)